# Exceptional water production yield enabled by batch-processed portable water harvester in semi-arid climate

He Shan [1,2], Chunfeng Li[2], Zhihui Chen[1,2], Wenjun Ying[2], Primož Poredoš[1,2], Zhanyu Ye[1,2], Quanwen Pan[2] ✉, Jiayun Wang [2] & Ruzhu Wang [1,2] ✉

Sorption-based atmospheric water harvesting has the potential to realize water production anytime, anywhere, but reaching a hundred-gram high water yield in semi-arid climates is still challenging, although state-of-the-art sorbents have been used. Here, we report a portable and modularized water harvester with scalable, low-cost, and lightweight LiCl-based hygroscopic composite (Li-SHC) sorbents. Li-SHC achieves water uptake capacity of 1.18, 1.79, and 2.93 g g$^{-1}$ at 15%, 30%, and 60% RH, respectively. Importantly, considering the large mismatch between water capture and release rates, a rationally designed batch processing mode is proposed to pursue maximum water yield in a single diurnal cycle. Together with the advanced thermal design, the water harvester shows an exceptional water yield of 311.69 g day$^{-1}$ and 1.09 g g$_{sorbent}$$^{-1}$ day$^{-1}$ in the semi-arid climate with the extremely low RH of ~15%, demonstrating the adaptability and possibility of achieving large-scale and reliable water production in real scenarios.

Freshwater scarcity has become a severe global challenge, since by 2025 almost 2 billion people will live in areas with absolute water scarcity[1,2]. A sustainable development goal raised by United Nations aims to solve water shortage with water availability and sustainability becoming of high concern[3]. Stable, portable, and high-yield clean water supply appears to be particularly vital, especially in emergencies or remote arid regions[4,5]. Atmospheric water, in the form of vapor and droplets, is a huge unexplored source of freshwater and its amount is equivalent to one-eighth of the total freshwater resources of rivers and lakes[6]. Harvesting overlooked but ubiquitous water resources in the atmosphere is considered as a potential approach to alleviate the water shortage[7]. Among all atmospheric water harvesting (AWH) technologies, sorption-based AWH (SAWH) shows high adaptivity and fewer limitations on environmental conditions, as sorbents have a strong affinity to capture water from air at a wide range of RH (~10–100% RH) and release it due to water vapor partial pressure difference[8,9].

The performance of SAWH systems highly depends on the water sorption capacity of sorbents. Recently, novel sorbents with high water uptake, low energy demand, fast kinetics, and cycling stability have been widely developed to improve the working performance of AWH systems[10], including metal-organic frameworks (MOFs)[11–16], hydrogels[17–22], and salt-based composite sorbents[23–28]. Salt-based composite sorbents are developed by inserting hygroscopic salt crystals into the porous matrix. The commonly used hygroscopic salts include lithium chloride (LiCl) and calcium chloride (CaCl$_2$), which show strong hygroscopicity and high equilibrium water uptake at low RH. Highly porous or capillary materials, such as fibrous substrates, polymer networks, and hollow containers, are frequently used as a matrix[10]. Unlike MOFs and hydrogels with relatively limited sorption humidity ranges, salt-based composite sorbents can realize excellent water uptake capacity at a wide range of RHs, thereby demonstrating a strong environmental adaptation. Besides, the synthesis of MOFs and hydrogels requires expensive raw materials, complex procedures, and strict reaction conditions, such as high temperature and pressure conditions of the hydrothermal reactions and the deoxidizing condition of free radical initiation synthesis, but salt-based composite sorbents can usually be easily synthesized by

[1]Institute of Refrigeration and Cryogenics, Shanghai Jiao Tong University, Shanghai 200240, China. [2]Engineering Research Center of Solar Power & Refrigeration, MOE China, Shanghai 200240, China. ✉e-mail: sailote@sjtu.edu.cn; rzwang@sjtu.edu.cn

impregnating matrixes with salt solution at mild ambient conditions. Therefore, the cheap and industrially mass-produced raw materials and the simple production processes make the salt-based composite sorbents highly suitable for mass production and show great market potential.

It is of great concern that although many high-performance sorbents have been reported, the practical photothermal water harvesters are still very rudimentary or small-scale, with simple heat and mass transfer solutions, resulting in low efficiency and limited overall water yield[29,30]. Recently, many approaches, such as thermal insulation by silica aerogels[14], selective solar absorbers[31,32], dual-stage latent heat recovery[31], and multi-cyclic operation modes[12,23,33] have demonstrated the improvement of the overall energy efficiency and the water yield of solar-thermal water harvesters. For example, multiple water harvesting cycles over the whole day were implemented in a MOFs water harvester, achieving the water productivity of 0.7 L $kg_{MOF}^{-1}$ $day^{-1}$ in the actual desert climate. Besides, the water harvester with daily eight water capture-release cycles recently achieved ultrahigh outdoor water productivity of 1.05 $L_{water}$ $kg_{sorbent}^{-1}$ $day^{-1}$ only driven by natural sunlight. Furthermore, combining AWH systems with other applications, such as thermal management[34–38], energy generations or storage[39–44], and agriculture[19,35,45] can also be regarded as the methods to improve the overall production or efficiency of the AWH-related systems. With these approaches, the practical outdoor water productivity per kilogram of sorbent has improved from 0.07 to over 1 L $kg^{-1}$ $day^{-1}$. However, challenges regarding high yield and stable water production remains. For passive photothermal water harvesters, the actual total daily water yield is limited to tens of grams, and they are overly sensitive to the fluctuation of solar energy, making it harder to meet the personal demand for daily drinking water. Contrary, active sorption-based water harvesters that rely on electric or thermal energy-driven sorption processes as input are getting more attention owing to their stable water productivity and significantly higher yield, whereas the weight and size of these harvesters are usually unacceptable, especially for portable SAWH applications[46,47]. This raises a question about the successful realization of portable, stable, and high-yield water harvesters that can meet the needs of emergency drinking water in multiple scenarios.

In this work, we introduce a portable and modularized water harvester with scalable, low-cost, and lightweight LiCl-based hygroscopic composite (Li-SHC) sorbents to realize a hundred-gram water production yield in a real semi-arid climate. Li-SHC shows excellent equilibrium water uptake and optimized sorption dynamics, achieving 1.18, 1.79, and 2.93 g $g^{-1}$ at 15%, 30%, and 60% RH within night-time 12 h. More importantly, the gaps between material-level water uptake and real device-level water production are noticed. Considering the mismatch between relatively low water sorption and fast desorption rates, a batch-processed operation strategy was proposed in which multiple pieces of sorbents simultaneously capture water vapor to make full use of the nighttime high RH and release them alternately to maintain the high desorption rate during the entire daytime. The combination guarantees high water uptake of each piece of the sorbent as well as high water production yield in a single diurnal cycle. Together with the device-level advanced thermal designs, the portable water harvester with the volume of 5.6 L and the weight of 3.2 kg achieved water productivity of up to 311.69 g $day^{-1}$ and 1.09 g $g_{sorbent}^{-1}$ $day^{-1}$, by applying eight desorption cycles in the real semi-arid environment (Lanzhou, China) with the extreme low RH of ~15%. Moreover, the harvester with lightweight sorbents can be easily deployed by a single person. With these features, the developed portable water harvester can realize high-yield freshwater production in real-world scenarios without the limitation of weather conditions and geographical locations. This study is expected to inspire further research focusing on the practical AWH applications to match actual human water consumption anytime and anywhere.

## Results

### Synthesis and characterization of Li-SHC

The sorbent Li-SHC was prepared by impregnating LiCl salt on active carbon felts which has been recognized as an excellent porous matrix choice for hygroscopic salts loading. The abundant micropores of carbon fibers provide ultra-high specific surface areas (Supplementary Fig. 1). Meanwhile, their physical entanglement and numerous channels enhance the hydrophilicity (Supplementary Fig. 2) and capillary force of the felt (Supplementary Fig. 3). The water uptake isotherm of the pure porous matrix (Supplementary Fig. 4) demonstrates the micropore filling at the low RH (<30%) and an increased water uptake at higher RH owing to the multilayer adsorption and capillary condensation in mesoporous[48], but the relatively low water adsorption capacity (<0.1 g $g^{-1}$ below 30% RH) can hardly be suitable for water harvesting in arid climates. Therefore, hygroscopic nanoscale LiCl salts were uniformly loaded to the porous matrix by vacuum impregnation and mild heating methods to promote the water harvesting capacity in a wide working range of RHs. Meanwhile, the active carbon fiber felt serves as the matrix to support and disperse LiCl to alleviate the salt agglomeration and salt solution leakage. After LiCl loading, the dried sorbent becomes less flexible and shows a higher mechanical strength than the pure matrix, but the flexibility of the sorbent can be recovered after sorption (Supplementary Fig. 5).

The successful LiCl salt loading and its hydrated salt crystallization form (LiCl and LiCl·$H_2O$) at different temperatures (30, 60, 90, 110 °C) were confirmed by powder X-Ray diffraction (PXRD) of Li-SHC (Fig. 1a). The cut-section scanning electron microscope (SEM) image (Fig. 1b) and the energy-dispersive X-ray spectroscopy (EDX) mappings of Li-SHC (Fig. 1c) confirmed the LiCl crystals were successfully loaded on the surface of fibers and between the channels formed by the physical entanglement. When exposed to moist air, the water molecules can be absorbed by LiCl crystals on the surface of the fibers (Fig. 1d), then diffuse into the porous matrix, forming concentrated LiCl solution. The volume of LiCl solution is expanded during the absorption process, and the formed solution is stored inside the hydrophilic and highly porous matrix with strong capillary force. A water vapor breathable but waterproof porous polytetrafluoroethylene membrane was further applied to encapsulate sorbents to completely avoid the risk of solution leakage and the resulting corrosion and cyclic sorption performance degradation (Supplementary Figs. 6–9)[32,49,50].

The hygroscopic LiCl and its content are the critical parameters determining water vapor sorption performance for water uptake capacity and dynamics. Dynamic vapor sorption curves of Li-SHC (Fig. 1e) with five different salt contents (40, 70, 80, 90, and 95 wt%) were measured to study the effect of the porous matrix and LiCl loading content. The linear drive force (LDF) model was used to quantitively evaluate their dynamic sorption characteristics (Supplementary Information Section 2)[51]. The sorption rate coefficients $k_{LDF}$ indicates a strong positive correlation between the salt content and the sorption rate (Supplementary Fig. 10). This variation of dynamic sorption characteristic owes to the mass transfer enhancement effect, provided by the matrix, as it leverages the effect of additional pores and channels for increased sorption interfacial area. However, with the increased salt content, LiCl crystals form not only on the fiber pores but also on the surface and between the gaps of fibers (SEM images see Supplementary Fig. 11), which inevitably reduces the vapor transport channels and the sorption dynamics. Note that although reducing the salt loading can increase the sorption dynamics, the absolute amount of water uptake decreases simultaneously. The desired salt content in practical water harvesting applications can be optimized by matching the duration of a sorption cycle with the required equilibrium duration, indicating no waste of both the sorbent material and the operating time[52]. Considering the 12-h nighttime sorption duration and performance degradation of scale-up sorbents in real climates, Li-SHC

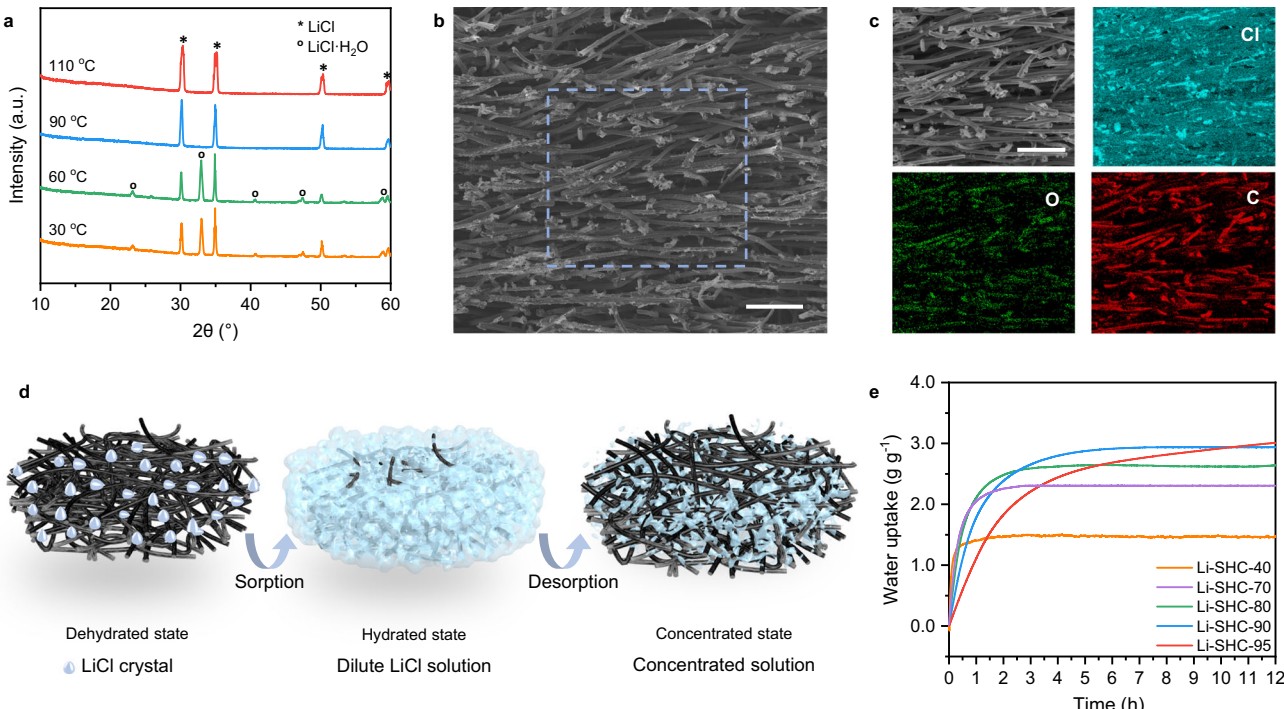

**Fig. 1 | Characteristics and optimization of Li-SHC. a** Variable temperature XRD patterns of porous matrix and sorbents. **b** SEM images of Li-SHC with the salt content of 90 wt% in the sectional view. Scale bar: 100 μm. **c** EDS element mapping of Li-SHC demonstrates the uniform nanoscale arrangements of LiCl crystals and small amounts of agglomeration. Scale bar: 100 μm **d** Schematic illustration of the sorption-desorption process. The water molecules are captured by LiCl crystals on the surface of the fibers, then diffuse into the porous matrix, forming concentrated LiCl solution. The solution is stored inside the hydrophilic and highly porous matrix with strong capillary force. **e** Dynamic sorption curves of Li-SHC with the salt content of 40%, 70%, 80%, 90%, and 95%.

with the salt content of ~90 wt% (referred to Li-SHC) was selected due to the best trade-off between sorption capacity and sorption dynamics.

The sorption isotherm of Li-SHC at 30 °C was shown in Fig. 2a. To reveal the sorption mechanism of Li-SHC, the phase diagram and the theoretical isotherm of pure LiCl were calculated (Supplementary Information Section 3) and presented in Supplementary Fig. 12 [53]. The isotherm of Li-SHC shows a highly similar isotherm of pure LiCl, which indicates that the equilibrium water uptake of Li-SHC is mainly determined by the sorption capacity of LiCl salt. In other words, the porous matrix has a negligible contribution to the equilibrium water uptake. The water uptake of Li-SHC shows the multi-step mechanism as demonstrated in Fig. 2a: 1. Chemisorption of anhydrous LiCl below the deliquescence RH (11% RH at 30 °C); 2. Deliquescence of hydrous LiCl (LiCl·$H_2O$) and the saturated solution forms 3. Absorption of water vapor in saturated/concentrated LiCl solution [53]. Note that over 85% of total water uptake was attributed by the adsorption of water vapor in LiCl solution when sorption at RH > 60%, thus the third step highly contributes to the overall water uptake capacity.

Owing to the ultra-high LiCl content of Li-SHC, the water uptake reaches 1.18 g g$^{-1}$, 1.79 g g$^{-1}$, and 2.93 g g$^{-1}$, when the RH is increased to 15%, 30%, and, 60% RH, respectively. The results of dynamic sorption experiments under three typical climates, arid climate (30% RH), semi-arid climate (60% RH), and humid climate (90% RH) demonstrate the environmental adaptivity of Li-SHC (Fig. 2b). Surprisingly, Li-SHC takes up more than 846% water content under humid climates (90% RH) within 12 h, revealing the ultra-high water uptake capacity of Li-SHC. In arid climates (30% RH), Li-SHC also shows satisfactory water uptake performance in comparison to other reported AWH sorbents (Fig. 2c and Supplementary Fig. 13)[1,11,12,18–20,22,23,25,26,28]. Besides, the satisfied sorption dynamics can also be observed because Li-SHC can reach equilibrium within six hours at the RHs of 30% and 60%. Moreover, the sorption capacity of Li-SHC is insensitive to the ambient temperature

because of the essential sorption mechanism of LiCl salt, thus it can be extended to extreme conditions regardless of air temperature (Supplementary Fig. 14). Sorption-desorption cycles performed over 186 h confirmed the stable equilibrium and dynamic sorption performance without degradation and LiCl leakage (Fig. 2d and Supplementary Fig. 15).

One of the most critical parameters for the evaluation of sorbent performance in real-world water harvesting is its sorption and desorption dynamics[54]. Based on a myriad of publications, it has long been ignored and less comprehensively evaluated, but with the development of the batch processing or continuous AWH, the importance of this parameter should be intensely evaluated and considered as well[55–57]. The reported sorption dynamics in most papers, including those presented in Fig. 2e, are usually conducted using milligram-scale samples inside an ideal constant humidity and temperature chamber with a fast air flow[58]. Although this thermogravimetric analysis is regarded as an appropriate scientific method for evaluating ideal sorption performance by controlling variables that might affect analysis, these results typically do not reflect the actual sorption state in practical climates and AWH devices because the performance of scale-up gram-scale sorbents is easily limited by higher vapor diffusion barriers and lower heat transfer rate, which is especially true for the monolithic sorbents with high packing thickness and density. Therefore, to evaluate the practical sorption performance in a real semi-arid climate more accurately, a massive sorbent (250 × 250 x 2 mm), whose weight is nearly 10000 times than those usually used in commercial sorption analyzer, was selected and tested in a typical semi-arid climate with the typical night-time humidity and temperature of 65% RH and 15 °C. The dynamic sorption curve under these nonideal conditions indicated, as expected, the slower water uptake dynamics (Fig. 2f). Note that this decrease in sorption performance is a common problem of sorbent materials used in devices, as even MOFs demonstrated a more serious performance reduction between milligram-

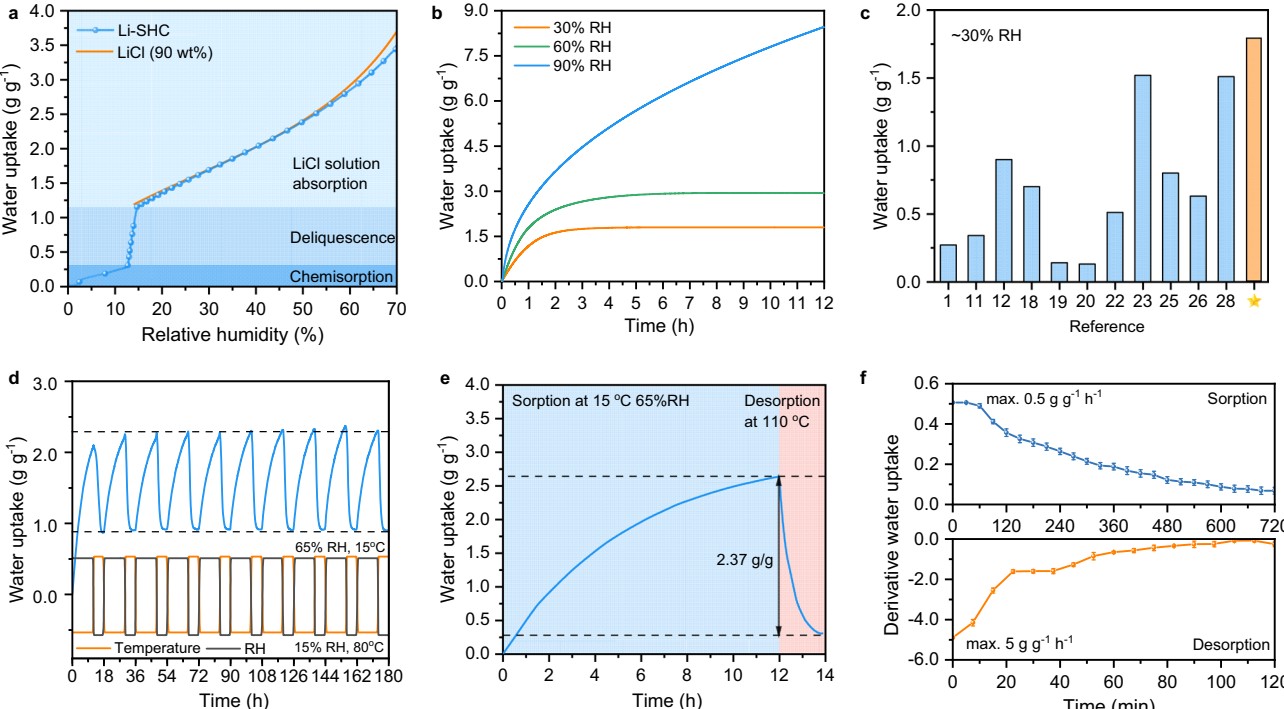

**Fig. 2 | Sorption characteristic and comparison of Li-SHC. a** Vapor sorption isotherm of Li-SHC and theoretical isotherm of LiCl with the initial specific mass of 90 wt%. **b** Dynamic sorption curves of Li-SHC at 30% RH, 60% RH, and 90% RH. **c** Comparison of vapor sorption capacity with the state-of-the-art sorbent materials at 30% RH. **d** Cycling performance of 2-mm-thick bulk Li-SHC. The sorption and desorption are performed at 65% RH, 15 °C, and 15% RH, 80 °C, respectively. **e** Dynamic sorption and desorption curves of 2-mm-thick bulk Li-SHC sample. The sorption curve was measured under a simulated semi-arid climate with the slight wind (15 °C, 65% RH). The desorption curve was measured using a hot plate at 110 °C. **f** Derivative water uptake changes of 2-mm-thick bulk Li-SHC sample demonstrating the mismatch between sorption and desorption time. Error bar: standard deviation (SD).

scale samples and bulk sorbents[33]. Furthermore, we measured the dynamic desorption performance of sorbents under a hot plate at 110 °C instead of placing them inside the constant climate chamber under ideal conditions. By comparing the derivative water uptake of sorption and desorption curves, we found a significant mismatch between water capture and release rates. The peak derivative sorption mass change is ~0.5 g g$^{-1}$ h$^{-1}$, only 1/10 compared with the results of the desorption condition (~5 g g$^{-1}$ h$^{-1}$). Therefore, two mismatches need to be particularly considered to pursue maximum water production yield if applying the sorbent to real devices: 1. the sorption performance of real bulk sorbents versus the milligram-level test samples; 2. the slow water sorption rate and the relative fast desorption rate. Fortunately, although a slower sorption dynamic limits the sorption performance of Li-SHC within 12 h, Li-SHC still shows a large amount of water release (2.37 g g$^{-1}$) in one water harvesting cycle under this actual situation, which benefits from the strong water affinity of LiCl salts, the high salt content of sorbents, and the better thermal conductivity of the carbon-based matrix. Also, to ensure quasi-continuous water collection during the daytime, a batch-processed operation mode is strongly needed to maintain a fast desorption rate during the entire daytime, which is expected to improve diurnal water collection rates.

## Design and optimization of water harvester

Enabled by the high-performance, adaptable and stable sorbent Li-SHC, a portable and modularized water harvester was designed, manufactured, and assembled to demonstrate the water harvesting potential (Fig. 3a-b and Supplementary Figs. 16–18). The water harvester includes an electrical heating plate hosting layer(s) of Li-SHC sorbents with a surface area of 25 × 25 cm². The heating plate required a 12 V DC power supply, which can be easily realized by the photovoltaic (PV) or PV-battery power supply (Supplementary Fig. 19). The temperature of the heating plate was controlled by a well-designed feedback system, which can accurately adjust the heating power and consequently temperature within 2.5 °C by applying the feedback control method (Supplementary Fig. 20). During the desorption process, a condensation cover was connected and sealed with the heating plate by a buckle structure, which can be easily dismounted and reassembled. The hot moist air was generated by applying heat to the sorbents and transferred to the condensation cover due to the concentration and density differences between the desorption and condensation parts. It was then condensed into liquid water droplets and slid into water reservoirs at the edges of the condensation cover. During the night-time sorption process, sorbents were placed inside the device, capturing water vapor from the air. In the daytime, sorbents with high water content were used to generate the water vapor through desorption.

The optimum heating temperature was around 110 °C for sorbent regeneration, which ensured the long-term cyclic performance of a sorbent. Generally, to improve the vapor condensation performance, it is desirable to increase the temperature difference between the sorbent and condensation parts[59], which is determined by the heat and mass transfer inside the chamber (Fig. 3c). To do so, temperature and velocity distributions inside the harvester were numerically simulated by the COMSOL software (Supplementary Information Section 5). The simulation results indicate a large air velocity in the unblocked cavity region, resulting in a fully developed natural convection of heated moist air (Fig. 3d). This leads to a higher condensation temperature, thereby decreasing the temperature gradient between the condensation cover and the heating plate, resulting in further reduction of liquid water generation. To alleviate this problem, a highly reflective heat insulation panel, made of polystyrene foam coated with aluminum foil, was placed in the space above the sorbents (Supplementary Fig. 21). According to the further simulation results, the natural convection and the direct thermal radiation between the condensation surface and

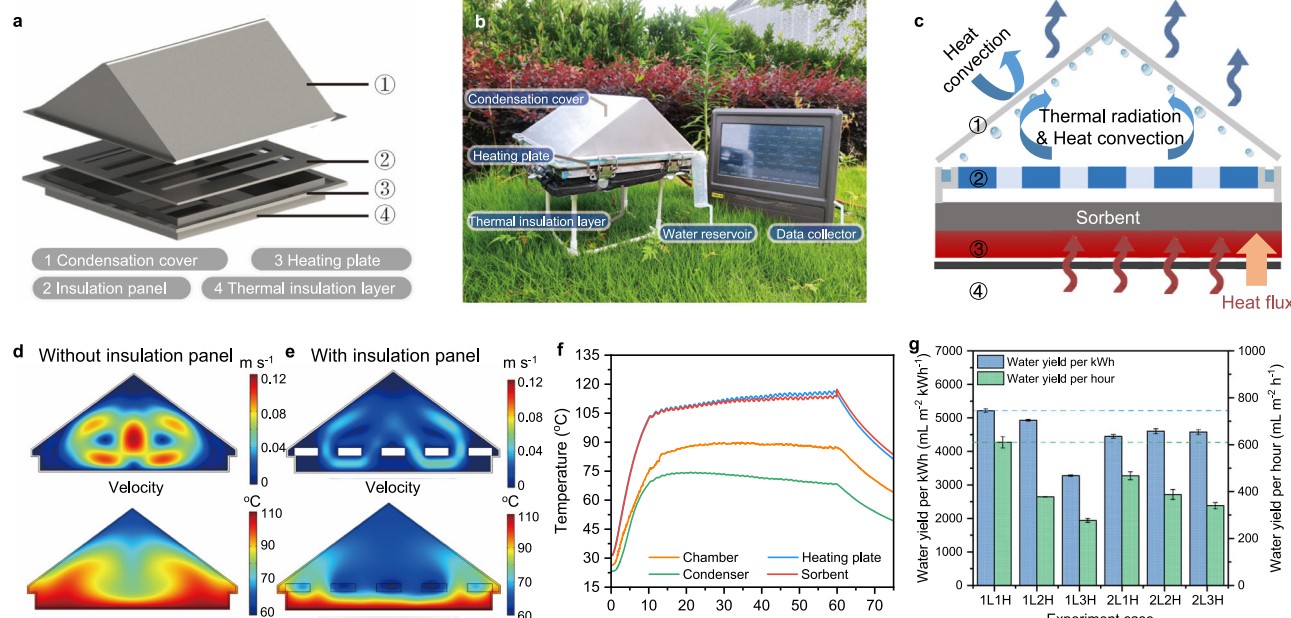

**Fig. 3 | Design and optimization of sorption-based water harvester. a** 3D render of the designed portable sorption-based water harvester. **b** Photo of the water harvester and the data collector. **c** Energy balance of the SAWH device during the water release-condensation process. **d** Velocity and temperature distribution inside the device without the insulation panel. **e** Velocity and temperature distribution inside the device with the insulation panel. **f** Temperature variation of the chamber, heating plate, condenser, and sorbent in the case of the 1-layer-sorbent and 1-heating-hour experiment. **g** Comparison of water yield per kilowatt-hour and per hour for six different experiment cases. L-layer(s), H-hour(s). Error bar: SD.

sorbents were remarkably reduced after installing this panel (Fig. 3e). The moist air movement is massively suppressed to the region below the insulation panel, which is exhibited by a highly increased temperature gradient. The steady-state temperature difference between the condensation surface and the heating plate reaches ~42 °C. To show the improvements after installing the panel more directly, the temperature variations of two selected characteristic points before and after installation were compared, showing a 25% drop in temperature (Supplementary Fig. 22).

Furthermore, we experimentally measured the temperature variations of the sorbents, chamber, heating plate, and condensation cover during the regeneration and collection experiment. As shown in Fig. 3f, the equilibrium-state surface temperature of the one-layer sorbent can reach up to 106.8 °C within 15 min under the heating temperature of 107.1 °C. Owing to the accuracy of the temperature control system, they are nearly stable during the following desorption process. Besides, the temperature difference between the sorbent and the heating plate was within 3 °C, proving the sufficient heat transfer performance of the sorbent. More importantly, as anticipated from simulation results, the temperature difference of 33–49 °C between the sorbent and condensation cover was achieved, which provided a strong driving force for the condensation process, ensuring a high liquid water collection rate.

To fully realize the daily water production potential of the water harvester, in addition to advanced heat and mass transfer study, we also investigated the optimization parameters and strategies. The heating hours (1–3 h) and the layer of sorbents (1 or 2) were selected as two optimization parameters to obtain maximum water productivity and minimize unit energy consumption. The parameter optimization experiments were conducted in the real semi-arid region (Lanzhou, China, 36.017° N, 103.784° E). Details information can be found in Supplementary Fig. 23–26. The results show that the total water yield increased from 609.6 mL m$^{-2}$ to 828.8 mL m$^{-2}$ with the increased heating time (Supplementary Fig. 27), but the water amount that produced in the single hour decreased by 55% in the case of one-layer

sorbent. This is consistent with the dynamic desorption result of sorbents (Fig. 2f), in which 78% of water was released within the first 1 h, whereas only 11% of water was released over the following 1 h. It further demonstrates that regenerating multiple pieces of sorbents successively each hour to fully maintain the fastest desorption rate of each sorbent can obtain the highest possible amount of water in a limited time.

The energy consumption of water production is another assessment criterion. Energy input determines the maximum theoretical amount of water production, especially when solar PV electricity is used for heat supply. As shown in Fig. 3g, the maximum water yield per unit of energy consumption (5215.1 mL m$^{-2}$ kWh$^{-1}$) also occurred in the case of the one-layer one-hour case. As for the one-layer three-hour case, the water yield per kWh dramatically declines to only 3274.4 mL m$^{-2}$ kWh$^{-1}$. It has been proven that when the heating time increased from two to three hours, more input energy cannot be used for vapor generation because at this time the sorbent has already reached the desorption equilibrium. To sum up, the 1-layer 1-hour case is the optimal mode to pursue higher water production and lower energy consumption within a limited time.

## Field test of batch-processed water harvester in semi-arid climate

The operation mode of the most reported SAWH devices is quite simple−single water capture-release cycle in a diurnal cycle, resulting in the inefficient water production, largely because of the apparent mismatch between the sorption and desorption dynamics. Such an operation strategy leads to inefficient utilization of solar energy and consequently lowers the corresponding water yield per 24 h due to the fast desorption rate with a depleted source of sorbed water (Supplementary Fig. 28a). Many research studies attempted to solve this mismatch by semi-continuous or continuous AWH cycles, but this means that the water capture cycles have to take place during the daytime at low RH environments (Supplementary Fig. 28b)[23,25,33]. However, the high RH brought in the nighttime due to diurnal air

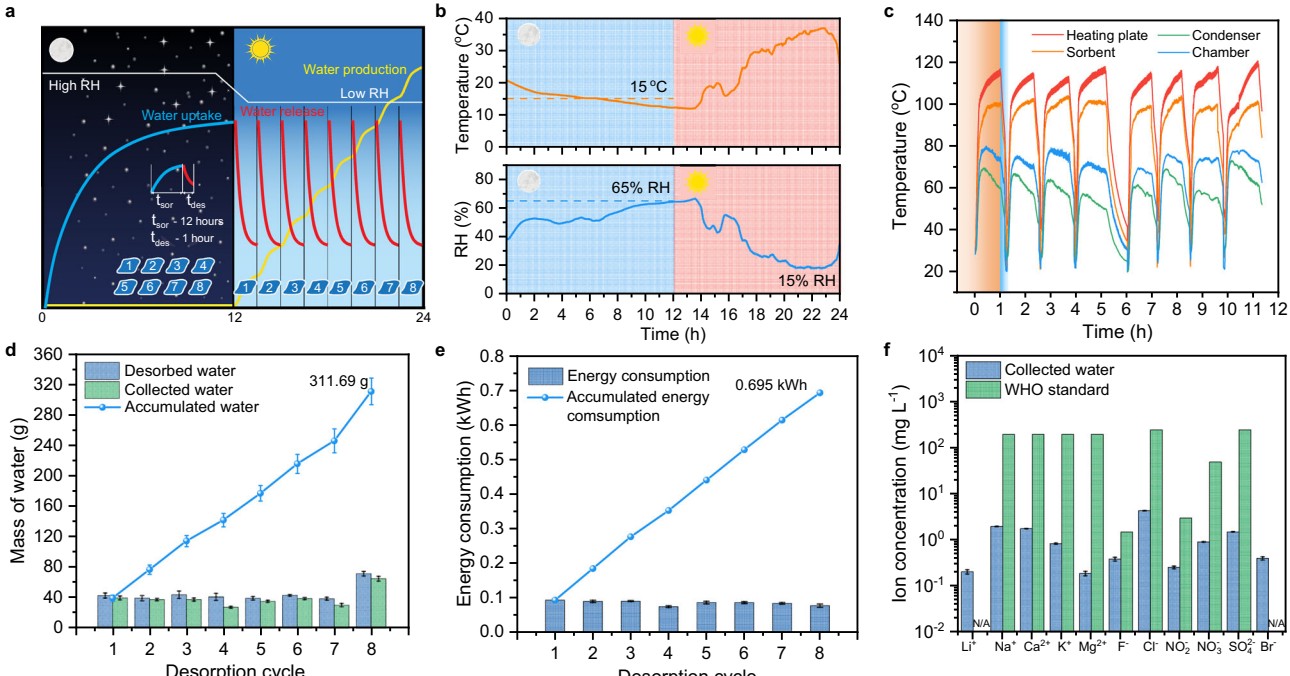

**Fig. 4 | Water harvesting performance of the water harvester by applying the batch-processed alternating mode. a** Schematic diagram of the proposed batch-processed alternating mode, showing the variation of water capture, release, and production in a single diurnal cycle. The eight pieces of sorbents are simultaneously exposed to the high RH ambient during the night, and consequent batch-operation desorption releases water vapor with freshwater production during the daytime. The white, blue, red, and yellow lines represent RH variation, water uptake, water release, and water production over the whole day, respectively. **b** Air temperature and RH variation within 24 h in the semi-arid region of Lanzhou, China. RH: relative humidity; **c** Temperature variations of the sorbent and device during eight desorption cycles including the heating stages and standby stages. **d** Mass of desorbed and collected water during each cycle, and its accumulated water. **e** Energy consumption during each desorption cycle. **f** Detected concentrations of the possible metal and ions in the collected water. All error bars represent SD.

temperature variation in the semi-arid climates, is more beneficial for the water sorption. To bridge this gap, a distinctive operation strategy, in which the multiple pieces of sorbents are simultaneously exposed to the ambient with high RH in the nighttime to absorb water vapor, then batch-processed alternately for water release during the daytime, is proposed to make full use of both the nighttime high RH environment and maintain high desorption rates throughout the day (Fig. 4a). In practical device operation, this operation strategy requires the portability, adaptability, and stability of both hygroscopic materials and water harvesters. The lightweight sorbent Li-SHC produced with low-cost and sustainable raw materials, facile and easily scaled-up fabrication procedures show excellent water sorption performance, adaptability, and stability, which allows fast deployment into practical applications (Supplementary Information Section 6). Additionally, the portable and cheap water harvester can be easily disassembled, re-assembled, and carried by a person. These features make it possible to realize ultra-high freshwater production yield in a diurnal cycle in real arid scenarios.

Based on the presented strategy, we demonstrate the viability of the multicycle mode and pursue its potential of maximum water productivity in a single day under practical climatic conditions. Specifically, eight AWH cycles were obtained by the batch alternating operation strategy in one day (Fig. 4b and details information see Supplementary Section 7). The daytime RH in Lanzhou, China recorded extremely low values of ~15% RH, while it increased to ~65% RH during the nighttime as the air temperature dropped by 15 °C (Fig. 4b). The eight pieces of sorbents were exposed to the nighttime high RH ambient and absorbed water vapor, then they were moved to the water harvester to release water vapor one by one from 8 AM. Then, the condensation cover was assembled and then the water release process started. The sorbent temperature rose rapidly to the set temperature

and was maintained for 60 min. The condensation temperature was between 71 to 50 °C and dropped with the decrease of the desorption rate. Each cycle consisted of a 60-min heating stage and a ~20-min standby stage (Fig. 4c). The total duration of the 8-cycle experiment was ~11.3 h.

The quantity of collected and desorbed water for each cycle is shown in Fig. 4d and Supplementary Table 7. Remarkably, the overall water productivity was 311.69 g, even in the extreme semi-arid climate with the lowest RH of ~15%, which make the daily water productivity of the portable sorption-based water harvester into the order of 100 grams. The energy consumption for each cycle is displayed in Fig. 4e and Supplementary Table 8, in which the average energy consumption is lower than that of the single-cycle shown above, because of the heat capacity of the device. This competitive water productivity (311.69 g day$^{-1}$) and associated low costs (0.19 \$ L$^{-1}$, 448.5 mL kWh$^{-1}$) enable the large-scale AWH in real semi-arid regions, showcased by our portable device as one of the most promising approaches to overcome the challenges of water supply in emergencies and rural areas[60]. Furthermore, we measured the concentrations of the metals and ions in the collected water (Fig. 4f), and the results show that the quality of the water extracted from the air meets the requirements for drinking water quality set by the World Health Organization (WHO). Finally, the stability of the device was evaluated by conducting the 6-day water harvesting and batch-processed water production cycling experiments under various simulated climate conditions, further demonstrating the stability of the bulk sorbents and the strong environmental adaptability of the water harvester (Supplementary Fig. 29).

Except for the price of produced water, various tradeoffs between water productivity, cost, weight, and volume of the device need to be considered during the design of AWH systems. Additionally, there are

multifarious sorbents, structures, and energy sources involved in these devices. For our work, the water productivity per gram of sorbents per day is 1.09 g of obtained water per gram of sorbent per day ($g_{water}\ g_{sorbent}^{-1}\ day^{-1}$), which has taken into account the mass of the eight replaced sorbents. From the whole device level, the portable water harvester with the volume of 5.6 L and the weight of 3.2 kg achieved the water production yield of 311.69 g per day, showing the apparent superiority regarding the weight and space of the whole device (Supplementary Fig. 30). These advantages are highly attributed to the advanced thermal design of the device to avoid the use of complex auxiliary devices, and the proposed operation strategy to fully utilize the desorption time.

For a comprehensive assessment of the water production potential, we selected five typical climates across the globe, including arid, semi-arid, and humid climates, and obtained their daytime and nighttime average ambient temperature and RHs (Supplementary Information Section 10). Then, based on the collected performance data in field tests, the water production of each location was estimated and shown in Fig. 5a. The water production is significantly influenced by the ambient RH, as over 1000 mL $day^{-1}$ water production could be achieved in a relatively humid climate (e.g., Birmingham, GRB) but the lower limit of water production (150 mL $day^{-1}$) was reached in the aridest month of the Sahara Desert (Kharga, EGY). Furthermore, the global daily water production potential of the water harvester is also estimated according to the Dubibib-Astakhov (D-A) equation[61], the global average annual RH, and the collected performance data of the proposed water harvester. The water production potential of the water harvester is conservatively estimated at over 350 mL $day^{-1}$ in the majority of locations, only except for the Tibetan Plateau, North Africa, etc. (Fig. 5b).

Our developed water harvester with a facile operation strategy shows its impressive record-breaking metrics, demonstrating a near realizable high yield of portable water production anytime and anywhere. Featured with a low total capital investment of both the sorbent and device and low operation costs, the portable water harvester demonstrated its market potential, making it closer to marketization and industrialization.

## Discussion

We demonstrated a realization of a high-performance and portable water harvester with scalable, low-cost, and lightweight sorbent Li-SHC, which was comprehensibly developed from the perspective of materials, as well as advanced structure design and operation strategy. With the optimized LiCl content, Li-SHC shows the water uptake capacity of 1.18, 1.79, and 2.93 $g\ g^{-1}$ at 15%, 30%, and 60% RH,

respectively. The bulk sorbent Li-SHC shows a high water release of 2.37 $g\ g^{-1}$ in a single water capture-release cycle under a simulated practical semi-arid environment. By combining simulations and experimental approaches, the portable water harvester with the volume of 5.6 L and the weight of 3.2 kg was designed and optimized. Together with the proposed novel eight-cycle batch alternating desorption mode with maintaining the high water desorption rate during the daytime, the portable water harvester achieved an exceptional water production yield of 311.69 g $day^{-1}$ and 1.09 $g_{water}\ g_{sorbent}^{-1}\ day^{-1}$ in an entire semi-arid region with an extremely low RH of ~15%, which puts the daily water productivity of portable sorption-based water harvester on the hundred-grams scale. Remarkably, this portable device shows apparent superiority regarding the weight and space of the whole device, demonstrating exceptional performance considering these metrics.

We anticipate that this study could make a closer step to practical AWH to meet the daily personal water demand and could serve as an inspiration for future research work related to batch-operated practical AWH systems. Currently, the desorption heat is provided by joule heating. The main advantage of the proposed desorption mechanism based on electrical heating is to obviate the intermittence and periodicity of solar thermal heating to desorb and collect the adsorbed water as much as possible during the whole day. Considering the high correspondence between arid regions and solar-rich regions, the possibility of using solar PV systems to boost low-carbon water harvesters was estimated (Supplementary Information Section 11). By matching the energy demand of the above 8 water harvesting cycles and the performance of commercial PV modules under different weathers, we calculated the demand for solar panel areas, showing that solar panels with an area of 1.11–2.08 $m^2$ can meet the requirements of the water collection cycle shown above. Therefore, the thin-film solar cells that can be carried by a person could be used for portable water harvesting, and the silicon solar cells could power the scaled-up water harvester in areas without an electrical grid. Furthermore, new research avenues related to direct solar utilization in the form of PV/T panels or individual solar absorbers in conjunction with thermal storage should be investigated in detail, which could improve the overall system energy efficiency. More importantly, storing solar energy in electricity (e.g., batteries) and thermal heat (e.g., phase change materials) could expand the water release process into the night, realizing all-day atmospheric water harvesting. Finally, further advancements in tailor-made sorbents, where new materials with exceptional physical and chemical properties are combined into composite sorbents could pave the road for next-generation sorbents. For instance, joule-heating electrical components could serve as a nano/micro porous matrix, also

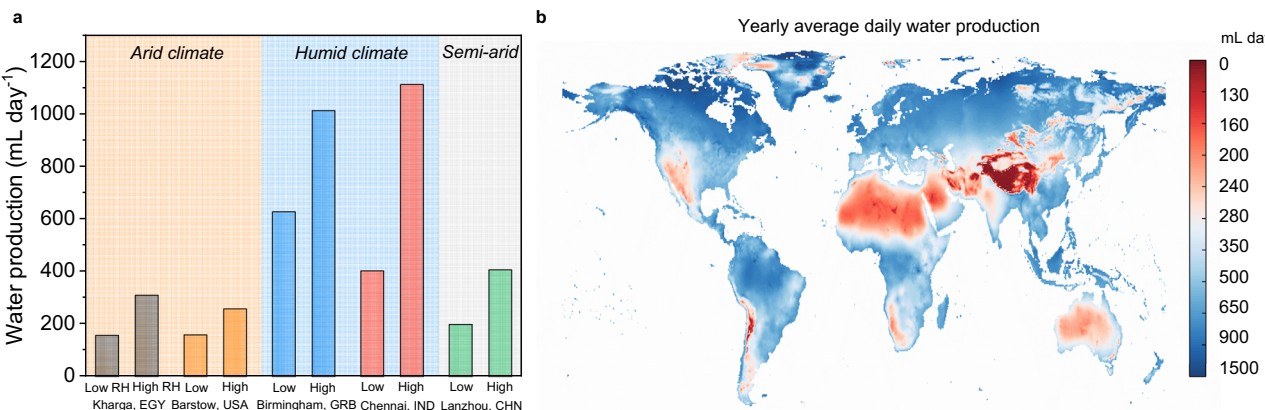

**Fig. 5 | Estimated water production of the water harvester applying the sorbent and operation strategy. a** Estimated daily water production in typical arid, humid, and semi-arid climates. The month with the lowest (highest) monthly average RH throughout the year is marked as a low (high) RH month. **b** Daily water production of the water harvester that is estimated by the yearly average RHs.

possessing high thermal conductivity, thus further reducing losses by introducing electrical heat utilization.

## Methods

### Synthesis and characterization of Li-SHC

Active carbon fiber felt was supplied by Kejing Carbon Materials. PTFE membranes were supplied by Zeyou Fluoroplastic. Lithium chloride (99%) was purchased from Sigma-Aldrich. Li-SHC were prepared by immersing the active carbon fiber felt pieces that were pre-dried at 120 °C for 8 h in pre-prepared LiCl solution with various concentration (0.05–0.45 g mL$^{-1}$). The wetted matrixes with the LiCl solution were then transferred into a vacuum drying chamber to eliminate the entrapped air in the matrixes. After 8 h, the wetted matrixes were taken out of the solution and then wrapped inside the porous PTFE membrane with a pore size of 3 μm. Finally, the sides of composite sorbents were encapsulated carefully to obtain the composite sorbents.

Nitrogen gas adsorption isotherms were recorded on volumetric gas adsorption analyzer at 77 K by 3Flex America Micromeritics. Attenuated total reflectance Fourier transform infrared (ATR-FTIR) spectra were recorded on the FTIR spectrometer (Nicolet 6700) using an attenuated total reflection (ATR) cell equipped with a Ge crystal. Powder x-ray diffraction (PXRD) patterns were recorded with a Bruker D8 ADVANCE diffractometer (Göbel-mirror monochromated Cu Kα1 radiation, λ = 1.54056 Å). The morphology and elemental distribution were examined by scanning electron microscopy (SEM) and energy-dispersive X-ray spectroscopy (Nova NanoSEM 230) equipped with energy disperse X-ray spectroscopy (EDS, X-MaxN 80, Oxford). The contact angles were measured by a surface contact angle meter (Data physics OCA20) at ambient temperature (-24 °C) using a 5 μL water droplet as the indicator. Inductively coupled plasma-optical emission spectroscopy (ICP-OES) (Avio 500) and Ion Chromatography (ICS-5000+/900) were used to assess the water quality.

### Measurements of water vapor sorption performance

Water sorption isotherms were measured by a commercial gas sorption apparatus (3Flex, Micromeritics). The dynamic water sorption–desorption tests of microgram-scale sorbents were performed on a thermogravimetric analyzer (STA 449C, Netzsch), equipped with a moisture humidity generator (MHG 32, ProUmid). The samples were completely dried at 120 °C, then placed in the thermogravimetric analyzer and kept at 30 °C under different humidity (30%, 60%, and 90% RH) for 12 h for water sorption.

Dynamic water vapor sorption-desorption experiments of the 2-mm-thickness sorbents were conducted in a constant climate chamber (KMF-115, Binder) with the temperature and RH accuracy of ±0.1 °C and ±1.5% RH. All samples were dried at 120 °C for 4 h to be dehydrated before the sorption tests. Once the chamber reached the set temperature (15 °C) and humidity (65% RH), the dehydrated composites were moved into the chamber. After a 12-hour sorption process, the sorbents were taken out and placed on a hot plate with a temperature of 110 °C to simulate the actual situation inside the AWH device. The mass change was measured by an analytical balance (Mettler Toledo ME204, 0.1 mg).

The sorption-desorption cyclic experiments were conducted to evaluate the stability of Li-SHC. The dehydrated sample with an area of 3 ×3 cm$^2$ firstly absorbed water vapor at 15 °C and 65% RH inside the constant climate chamber (KMF-115, Binder) for 12 h. After that, the environment was changed to the desorption condition (80 °C, 15% RH) for 6 h. The sorption-desorption cycles were repeated ten times, which took over 10,000 min. The weight change of the sample was recorded by the analytical balance. The stability and adaptability of the water harvester were evaluated by the 6-day water collection experiments under three various simulated climatic conditions. Detailed information can be found in Supplementary Information.

### Device fabrication and data acquisition

The water harvester was composed of an electrical heating plate (25 cm × 25 cm) attached to a sorbent container (29 cm × 29 cm × 2.4 cm) made of a 19 mm thick aluminum plate welded to the upper side of the side walls (25 cm × 1.9 cm). A roof-shaped condensation cover with an angle of 37° with the horizontal line connected with the sorbent container. A blind-style high reflectivity and heat insulation panel (254 mm × 254 mm) with four empty areas (25 mm × 190 mm), made of polystyrene foam coated with aluminum foil, was placed in the space above the sorbents. Detailed information on the water harvester, temperature control system, and heating system can be found in Supplementary Information.

As for the data acquisition system, an ultra-thin K-type thermocouple with a diameter of 0.1 mm (−20-200 °C, ±1 °C) was used to measure temperatures. The measure points were arranged on the surface of the heating plate, the inner surface of the condensation part, the central point of sorbents, and the vapor-flowing path. These signals were transferred to an Agilent 34970A. The current signal was transferred by the transmitter to the data acquisition system. All data were recorded and processed by PC.

### Field test in the semi-arid climate

The field tests of the water harvester were conducted in a real semi-arid environment, which is located in Xiagouya Mountain, Lanzhou, China (36.017° N, 103.784° E) in September 2021. The indoor and outdoor temperature and RH were recorded by two temperature and RH sensors (COS-03, Renke) with an accuracy of 0.15 °C and 1.5% RH.

The parameter optimization procedure before the batch-process experiments were conducted within two days. The dry weights of sorbents were between 37.90–39.65 g. All sorbents were exposed to the air for 12 h from 20:00 to 08:00 the next day. After that, the weight of the sorbent was recorded by a balance (0.01 g), and the water uptake of the sorbents was calculated (1.83–2.19 g g$^{-1}$). Then, the sorbents were placed inside the water harvester for water desorption and collection. The ambient temperature and RH during the desorption were 30 °C, and 20% RH, respectively. The device adjustment, batch-process multicycle experiment, and repeated experiments were conducted on the following days. The sorption process was from 20:00 to 08:00 during the night, and then the sorbents were alternately sealed and placed inside the water harvester. Taking the time lag of vapor condensation and the frequency of material replacement into account, each desorption duration was -1.35 h, and 8 cycles took around 11 h. The desorption time was from -08:00 to -19:30. Note that the fourth cycle was longer than the others. This rest time is in line with the special situation, such as an emergency water supply, that is operated by a single person, indicating that each AHW cycle could be decoupled and paused. The number of cycles can be determined by the daily water demand and the allowed operating time, and the maximum recommended number of cycles is eight.

## Data availability

All the data needed to evaluate the conclusions in the paper are present in the paper and/or the Supplementary information. Source data are provided with this paper.

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

## Acknowledgements

This research work was funded by the Foundation for Innovative Research Groups of the National Natural Science Foundation of China (No. 51521004). We thank Prof. Jinping Li and Mr. Rui Li at the Lanzhou University of Technology for their help.

## Author contributions

H.S., C.L. designed the prototype and synthesized the sorbents. H.S., Z.Y., and Z.C. carried out the experiments and analyzed the experimental data. W.Y., J.W. conducted the simulation. H.S., Q.P., P.P., R.W. prepared the manuscript. All authors contributed to the final version. R.W. and Q.P. conceived the idea and led the project.

## Competing interests

The authors declare no competing interests.
