## [Peer Review File · Nature Communications]

Exceptional Water Production Yield Enabled by Batch-process Portable Water Harvester in Semi-arid ClimateReviewers' Comments:

Reviewer #1:

Remarks to the Author:

This article named "Exceptional Water Production Yield Enabled by Batch-process Portable Water Harvester in Semi-arid Climates" shows a practical atmospheric water harvesting (AWH) path to achieve an exceptional water production yield.

The atmospheric water harvesting experiments were carried out in the real semi-arid area with an extremely low relative humidity, which could be a strong demonstration for practical AWH in these areas. An easy-to-synthesis hygroscopic material served as a foundation for water capture and release in a diurnal cycle. A portable harvester was designed and optimized using advanced thermal solutions. Besides, the rational batch-process operation could pursue maximum daily water production. With optimized thermal design and rational batch-process operation, the water harvester shows over hundred-gram daily water yield (312 g/day) in the limited volume (5.6 L) and weight (3.2 kg). To my knowledge, this is perhaps the 1st demonstration of a portable atmospheric water harvester.

Most importantly, different from the previous studies, this article focused not only on the materials design, but also on the improvement of thermal efficiency, the enhancement of mass transfer, and the real realization of high water yield in semi-arid areas. The developed material and water harvester can be easily scaled-up and reach the unprecedented water production yield, thus it may inspire further research on both the development of hygroscopic materials and the design and optimization of practical water harvesters. Therefore, I'd like to give my recommendation as acceptance after minor revisions.

1. It highlights that the synthesized hygroscopic material Li-SHC is with the characteristics of low-cost, easy-to-preparing, and easy-to-scale-up, which could be applied to the practical large-scale AWH. However, many intensive research articles reported these characteristics of their developed materials (such as MOFs and hydrogels). Compared with these materials, how is Li-SHC comparable in large-scale scenarios? How about the stability of bulk materials during the long-term application? Besides, the cost to synthesize the material needs to be analyzed in detail.
2. The thermal design for an AWH system is critical, there are many AWH devices that use fans as the mass transfer enhancement measures. Is this water harvester also applying such measures to boost mass or heat transfer? Besides, in Figure 2B, is there also a convective heat transfer (heat loss) between the heating plate and the ambient air? Are there some thermal insulations between the heating surface and ambient? If possible, please clarify it in the text.
3. The 8 times water release cycle was taken to pursue the maximum daily water yield. Is it possible or necessary to increase the number of cycles further? If there is only one water release cycle in limited duration, how about the water productivity of the portable device?

Reviewer #2:

Remarks to the Author:

The manuscript entitled "Exceptional Water Production Yield Enabled by Batch-process Portable Water Harvester in Semi-arid Climate" by Shan et al. shows the development of a portable and modularized sorption-based water harvester with ultra-high-water release in one water harvesting cycle under semi-arid climate. Moreover, Wang and his collaborators designed and optimized the portable water harvester by combining simulations and experimental methods. The presented article contains both basic experimental and application simulation tests, which are vital to support the claim with reliable data and in turn pave the way towards the application of this material in the field of water production in semi-arid regions.

In general, the work is accurate and well presented as a Nature Communications article should be, moreover the presented results are certainly of interest to readers of this journal. However, some

issues are supposed to be fully addressed to further improve the quality of this presented work.

1. In the abstract, "with ultra-high-salt-content sorbents up to 90 wt%" is mentioned, is this something that must be emphasized in the abstract? I think this should not be the focus and innovation of this work, so I suggest deleting it in the abstract.
2. Obviously, the material preparation in this study is very simple and does not seem to be innovative. The authors need to refine the unique advantages and innovations of this material in the manuscript.
3. In Line 66, "its production costs" should be changed to "their production costs".
4. In Line 124, the authors claim that "Since the porous.....water sorption". I believe it would be better to provide more references in order to support the conclusion by previously published articles in that field.
5. In Figure 1E, how to calculate normalized water uptake?
6. Does the super-high salt content make the material brittle? How about the mechanical properties of the prepared material? How about the cycling performance? Does the salt content of the material decrease after repeated use?
Generally, LiCl impregnated with materials is easy to lose. Is LiCl in the sorbent prepared in this paper easy to lose? How is LiCl preserved and stable and long-lasting in the material?
7. Authors claim that "A water vapor breathable but waterproof membrane...". How is the moisture permeability of PTFE membrane? Please supplement this data in Supplemental Information.
8. In Line 394, "2.37 g/g" needs to be bracketed.
9. In Figure 3B, why does the fourth cycle take longer than the others?
10. The use of "first" in the article should be avoided as much as possible. Please see Lines 332 and 401.
11. Figure 4B seems more appropriate to be placed in the Supplemental Information. Or the authors can find some supplementary interpretation of the figures 4A and 4B and combine them into new Figure 4.
12. Figures 1F, 1L, 3C, 3D and 3E report interesting quantitative data that are very useful to prove the final applicability of the developed material. Anyway, the bars representing the standard deviations should be included.

Reviewer #3:

Remarks to the Author:

The manuscript extensively describes an investigation aimed at developing a device for water harvesting from the atmosphere in dry diurnal climates such as in deserts. The investigation effort goes above and beyond, from material preparation all the way through device testing in a real setting.

The most noticeable outcomes of this research consist in the exceptionally high water working capacity and the record daily water yield (per gram of sorbent, per kilogram of device and per volume of device), which also constitute the two main claims of this research.

It is worth noticing that the results apparently stand out when compared with the following two publications:

- 1) Hanikel N, Prevot MS, Fathieh F, Kapustin EA, Lyu H, Wang H, et al. Rapid Cycling and Exceptional Yield in a Metal-Organic Framework Water Harvester. ACS Cent Sci. 2019;5(10):1699-706
- 2) Xu J, Li T, Yan T, Wu S, Wu M, Chao J, et al. Ultrahigh solar-driven atmospheric water production enabled by scalable rapid-cycling water harvester with vertically aligned nanocomposite sorbent. Energy & Environmental Science. 2021

The second reference above shares similarities with the manuscript in the salt used to impregnate the matrix (LiCl), in the research approach (also in this case the characterisation of the material was followed by the characterisation of the structure and final testing in a real environment) and in the performance achieved by the material (Figure S23).

This is not surprising since both the carbon-supported material of this manuscript and the carbon-

supported material of the second reference have essentially the same isotherm, which is the isotherm of the pure LiCl given that the composite material is impregnated with LiCl above 15% in weight. 15% is usually the boundary between a composite material with intermediate equilibrium properties between support and salt and a composite having the equilibrium properties of the salts, scaled down by the amount of salt impregnated.

LiCl salt has deliquescence at 11.3% Relative Humidity at 30°C that is exactly what Fig. 1F shows. The salt isotherm is not particularly sensitive to the temperature (nothing beyond the sensitivity of the relative humidity to temperature). In fact, increasing the temperature to 100°C, the deliquescence only moves to 9.9% Relative Humidity.

However, entering the conditions of Fig. 3A (adsorption at about 50% relative humidity average; desorption at about 20% relative humidity average) in the isotherm of Fig. 1F shows that in the real tests, the material worked in the salt solution zone. Besides, in a location where the day/night relative humidity swing is 30%, much more than in the location of Xu et al. *Energy & Environmental Science* (2021), so likely leading to better results.

Working in the salt solution zone makes wondering about the stability of the material. This important information is not present in the manuscript. The 8 samples of Fig. 3B have been used only once. Information whether or not the material can be used multiple times should be given. This would require also a better specification of the micropore size, whether crystals are also in the pores and their influence (likely negligible given the high loading) on the overall behaviour of the material. This raises also a question whether the carbon fibre support is really needed.

Another concern is about Fig. S23 and the closeness of the results achieved in this manuscript and those in Xu et al. *Energy & Environmental Science* (2021).

As already discussed, the similarity is due to the same LiCl for both materials. Fig. S24 shows an advantage of the device in the present manuscript that is due to the absence of the PV panel and of the fan, being the device of this manuscript less sophisticated (a no-fan and grid-connected, optional PV connection but not experimented).

However, the main factor contributing to the unrivalled amount of water produced is the periodic replacement of the sorption material. This is a practice common to both the present manuscript and of Xu et al. *Energy & Environmental Science* (2021), which is from the same senior author. All other compared systems do not use to replace manually the sorption material, hence achieving lower water yields.

Finally, there are some minor points that require the authors' attention:

- 1) The English shows sometimes missing "s", "these" instead of "this", "Contrary" instead of "Conversely", etc..
- 2) "Kinetics" is improperly used in the manuscript to refer to the dynamic behaviour of a sorbet structure. Kinetics is the diffusion in the material, which is a property of the material distinct from other properties such as the thermal conductivity for example. The authors should use the term dynamics in experiments in which the heat and mass transfer affect each other (typically when the amount of material is significant, usually in the order of mg or more).
- 3) Row 268 reports "Section S3" while it should be "Section S4"

Reviewer #1:

This article named "Exceptional Water Production Yield Enabled by Batch-process Portable Water Harvester in Semi-arid Climates" shows a practical atmospheric water harvesting (AWH) path to achieve an exceptional water production yield.

The atmospheric water harvesting experiments were carried out in the real semi-arid area with an extremely low relative humidity, which could be a strong demonstration for practical AWH in these areas. An easy-to-synthesis hygroscopic material served as a foundation for water capture and release in a diurnal cycle. A portable harvester was designed and optimized using advanced thermal solutions. Besides, the rational batch-process operation could pursue maximum daily water production. With optimized thermal design and rational batch-process operation, the water harvester shows over hundred-gram daily water yield (312 g/day) in the limited volume (5.6 L) and weight (3.2 kg). To my knowledge, this is perhaps the 1st demonstration of a portable atmospheric water harvester.

Response:

We really appreciate the reviewer for showing interest in our work. As you mentioned above, the highlight of our work is that the developed portable water harvester achieved exceptional water production in a real semi-arid region. The significant performance lift of freshwater generation was realized by the improvements regarding all important aspects of the AWH device from highperformance sorbent, to advanced thermal solutions of the device, and the batch-process operation strategy. And especially, we demonstrated the portability and stability of the water harvester in a real semi-arid region with the extremely low RH of 15%.

Most importantly, different from the previous studies, this article focused not only on the materials design, but also on the improvement of thermal efficiency, the enhancement of mass transfer, and the real realization of high water yield in semi-arid areas. The developed material and water harvester can be easily scaled-up and reach the unprecedented water production yield, thus it may inspire further research on both the development of hygroscopic materials and the design and optimization of practical water harvesters. Therefore, I'd like to give my recommendation as acceptance after minor revisions.

Response:

We thank the reviewer very much for seeing our scientific contribution in the field of AWH and recommending our work for publication. Indeed, our mission with this research study was to dive into uncharted territory by making a portable device producing the water on the hundred-gram scale in semi-arid regions by leveraging a comprehensive strategy, so it could inspire possible further research on the advanced designs on both material- and prototype-levels. We carefully considered all your valuable comments and provided a point-by-point response to the comments shown below.

1. It highlights that the synthesized hygroscopic material Li-SHC is with the characteristics of low-cost, easy-to-preparing, and easy-to-scale-up, which could be applied to the practical large-scale AWH. However, many intensive research articles reported these characteristics of their developed materials (such as MOFs and hydrogels). Compared with these materials, how is Li-SHC comparable in large-scale scenarios? How about the stability of bulk materials during the longterm application? Besides, the cost to synthesize the material needs to be analyzed in detail.

Response:

Thank you for your comments. We would like to comprehensively answer these questions from three aspects: the stability, cost, and water production of the sorbent and the water harvester.

Stability:

The stability of the material and device is indeed an important issue in a long-term application, and we comprehensively did the stability evaluations of both the device and the material.

Fig. R1 Stability evaluation of the sorbent. **a** 180-hour sorption-desorption cycling test of the sorbent Li-SHC at 65% RH, 15 °C for sorption and 15% RH, 80 °C for desorption. **b** Photos of the sorbent Li-SHC after exposure to the high RH of 90% RH, showing that solution leakage was avoided by the membrane encapsulation.

Fig. R2 Stability evaluation of the device at various simulated climate conditions. **a** Photos of the produced fresh water on each day. **b** Collected freshwater amounts of each cycle and each day. **c** Overall water production during the cyclic tests. The insert shows the photos of the overall collected water each day **d** Overall energy consumption during the cyclic tests. **e** Mass changes of each piece of the sorbent during the 6-day water capture-release cycling tests. **f** Temperature variation of the heating plate, sorbent, chamber, and two measuring points on the condenser. The measuring points of the sorbent moved due to released vapor turbulence, resulting in the measured temperature fluctuations.

To evaluate the stability of the sorbent, we performed a 180-hour water sorption-desorption cycling experiments using the sorbent with the area of 9 cm² inside the constant environmental chamber. The sorption and desorption condition are 65% RH, 15 °C and 15% RH, 80 °C, respectively. As shown in Fig. R1a, no obvious performance degradation was found during the sorption-desorption cycle. Note that the water uptake of the first sorption cycle is less than the remaining cycles as in the first active stage the sorbent absorbed water vapor from the total dry state. In addition, the risk of the salt solution was tested at high RH (75% RH and 95% RH), showing that no solution leakage was observed even at RH as high as 95% (Fig. R1b). It demonstrated the desirable stability of the sorbent Li-SHC, ensuring the reliability of the sorbents used in the batch-process mode.

To evaluate the stability of the device, we conducted a 6-day water harvesting and batch-processed water production cycling experiments under three typical climate conditions in our field test location, Lanzhou, China. The three nighttime sorption conditions are the typical climate in the spring/autumn season (15°C, 65% RH), the typical climate in the winter season (5°C, 50% RH), and the typical climate in the summer season (20°C, 75% RH). Experiments under each climate condition were conducted for 2 days, with a total experimental time of 6 days. As shown in Fig. R2a and Fig. R2b, the water production remained consistent in each climate condition, which are ca. 340 g at the spring condition, ca. 280 g at the winter condition, and ca. 425 g in the summer condition. The total water production over the 6 days is ~2115 g (Fig. R2c), showing the excellent water production performance and the adaptivity of the water harvester to different climates. As shown in Fig. R2d, the total energy consumption is 5.97 kWh, corresponding to the water production of ~355 g per kilowatt-hour.

In addition to the overall water production and energy consumption, the mass changes of the sorbents during the cycles are shown in Fig. R2e. It shows that the average water uptake of each sorbent is 2.41 g/g, 2.04 g/g, and 2.81 g/g for each climatic condition, respectively. More importantly, all sorbents have no obvious mass loss after these cycles, showing excellent stability throughout the cyclic tests with variable working conditions. Detailed information on the temperature variation of each component of the water harvester can be found in Fig. R2f.

Freshwater production assessment:

Fig. R3 Extensive techno-economic assessment of the water harvesters with the sorbents

As the Editor and Reviewer suggested, we conducted the techno-economic assessment to evaluate the cost and profit of producing freshwater by the water harvester in different regions and climates. The first part is related to the freshwater production assessment. Before that, we would like to emphasize that the minimum amount of water is ~ 4 L/day per person for basic personal survival, as recommended by the US National Academies of Science¹. If considering drinking water, sanitation, bathing, and cooking, it is estimated that an individual requires a minimum amount of freshwater equal to 50 L/day². Researchers in this field are still pursuing to reach the goal through both materials designs and system optimizations. Beyond that, the atmospheric water harvesting technology has shown the potential to change traditional approaches to access the water, especially when it is not easily accessible such as during disasters, emergencies and in regions experiencing water stress, where the liquid water is typically carried by individuals or transported from remote locations, accompanied by heavy costs. Meanwhile, these point-of-use case use scenarios using the off-the-grid AWH devices have the potential to provide safe water by obviating the need to build pipes, use water transportation using trucks or extract contaminated surface water³. Therefore, as shown in Fig. R3, the techno-economic assessment we conducted is referring to the AWH technology that can provide safe freshwater anytime and anywhere.

Fig. R4 Prediction of the sorption amount during the daytime under various conditions. **a** Experiment vapor sorption isotherm at 30 °C. **b** Calculated D-A fitting equation curve.

We estimate the water production of the water harvester at two levels. One is the global annual average water production, and the other is water production estimation during different seasons in five typical climates. Due to the variability of the sorption temperature and RH, we firstly established a model for predicting the water sorption amount of the sorbents under various sorption conditions. The Dubinin-Astakhov (D-A) equation was selected to predict the sorption capacity, which is derived from the experimental vapor sorption isotherm (Fig. R4a).⁴ The equation can be expressed as which can be express as:

$$w = w_{eq} \exp(k\Delta F)$$

where w is the equilibrium sorption capacity (g/g), and w_{eq} , k , n are the fitting coefficients. In the above equation, ΔF is the free sorption energy (i.e, sorption potential), which is the function of relative pressure (relative humidity) and the temperature:

$$\Delta F = RT \ln \frac{P}{P_0}$$

where R is the gas constant ($J \cdot mol^{-1} \cdot K^{-1}$), P , P_0 are the partial pressure and the saturated vapor pressure of water vapor (Pa) respectively. Therefore, various sorption condition (temperature and RH) at different locations and seasons can be expressed by only one parameter – the free sorption energy. The fitting equation results are shown in Fig. R4b and listed in Table R1.

Table R1 D-A sorption characteristic fitting curve of the sorbent

Stage	F (kJ/kg)		Correlation curves	R ²
I	1552.77-285.69	w_{eq}	$\exp(0.4146 - 0.00596\Delta F)$	0.9779
II	285.69-264.24	w_{eq}	$-0.041\Delta F + 11.99$	0.9907
III	264.24-29.49	w_{eq}	$22.63 \exp(-0.6471\Delta F^{0.2729})$	0.9996

After that, the global average annual temperature and RH data were obtained from Univ. of East Anglia and used together with the above equations to estimate the harvested water amount of the sorbent⁵. Then, the average water production is calculated based on the assumption of constant desorption and condensation rate that we obtained from the field tests. Meanwhile, due to the dispersion and nonuniformity of weather stations, a two-dimensional grid scatter interpolation method based on onedimensional periodic boundary conditions of longitude is used for interpolation. A map toolbox from MATLAB software was used to represent the water production map.

Fig. R5 Estimated annual daily water production that predicted by the global yearly average RH.

The estimated daily water production of the water harvester working in the batch-process mode is shown in Fig. R5. The results show that the global average annual water production is around 500 mL. The water harvester works in the Sahara Desert with ultra-low RH, yet it could still produce an average of 200 mL freshwater per day, showing excellent water sorption capacity of the sorbents and efficient freshwater production by the water harvester.

To be more specific, we selected five global climates globally with their corresponding monthly average ambient conditions (temperature and RH) which were obtained from the Energy Plus software, as shown in Fig. R6a-f. The five regions listing five typical climates were Birmingham, UK (humid climate and low ambient temperature), Chennai, India (humid climate and high ambient temperature), Kharga, Egypt (arid climate, near the Sahara Desert), Barstow, California (arid climate with high ambient temperature, near the Mojave Desert), Lanzhou, China (semi-arid climate with large seasonal temperature differences). Different from the global water production estimation using the yearly average RHs, the water production for the five typical climates used the nighttime temperature

and RHs (from 18:00 to 06:00 the next day), which improves the estimation accuracy. Besides, as shown in

Fig. R6a, we selected two typical months with the lowest or highest monthly average RHs in the whole year in each region. We believe these can further comprehensively evaluate the working performance of the water harvester in both different locations and seasons.

Fig. R6 Monthly average typical daily temperature and relative humidity in **a** Birmingham, UK; **b** Chennai, India; **c** Kharga, Egypt; **d** Barstow, California; **e** Lanzhou, China. Predicted daily water production in **f** Birmingham, UK; **g** Chennai, India; **h** Kharga, Egypt; **i** Barstow, California; **j** Lanzhou, China

The predicted water production (g/day) of each location and climate is shown in Fig. R6g-i. The results show that the production is significantly influenced by the ambient relative humidity. For instance, the production in relative humid regions (such as, Birmingham, UK and Chennai, India) was calculated to be over 1,000 g per day in high RH months. In contrast, the minimum water productions that can be achieved in the low RH reasons in desert arid climates (Kharga, Egypt and Barstow, California) was estimated at ca. 160 g per day, which could be considered as the minimum daily water production of the water harvester. All the water production estimations demonstrated the adaptivity and excellent working performance of the water harvester.

Cost:

Costs of the water harvester

The total costs of water production include the operating costs and the total capital investment.

Regarding the total capital investment, it consists of the total capital investment of the device and the costs of the sorbent, and each of them includes both raw materials costs and manufacturing costs.

Table R2 Total capital investment of the device (unit: \$USD)

Raw materials	Price of one prototype	Price in case of mass production	Manufacturing process	Price of one prototype	Price in case of mass production
Stainless steel	23.6	11.0	Cutting & welding	94.5	32.5
Thermal insulation foam	1.6	1.6	Assembly	7.9	7.9
Temperature-control system	34.7	28.4	Polishing	10.0	5.0
Buckles	18.9	12.6			
Total	78.8	53.6	Total	112.4	45.4

The cost assessment of the water harvester is listed in Table R2, showing the total price of one prototype is \$USD 191.2, and if it is mass-produced, the price could be reduced to less than \$USD 100. The costs of the raw materials of the water harvester include the stainless steel, thermal insulation foam, buckles to seal the device, and the temperature-control system. The most expensive part is the temperature-control system (\$28.4-34.7). Note that the raw materials required for the fabrication of the device are common and can be mass-produced, which further reduces the costs of the device. Besides, the plastic buckles and aluminum materials can be used to further control the costs instead of using more expensive stainless steel and metallic buckles. Regarding the manufacturing costs, the price of the manufacturing process can be divided into the depreciation costs of the equipment and the costs of labor-hour, which are more expensive than the raw materials.

Costs of the sorbent

Table R3 Total cost of the sorbent (unit: \$USD)

Raw materials	Price of one prototype	Price in case of mass production	Preparation process	Depreciation expense of the equipment	Specific energy consumption
Lithium chloride	6.4	0.3	Drying	0.04	0.42
Active carbon fiber felt	0.5	0.4	Stirring	0.03	0.16
PTFE porous membrane	0.4	0.3			

Total	7.2	1.0	Total	0.65
-------	-----	-----	-------	------

The total cost of the sorbent is listed in Table R3, and the costs of both raw materials and preparation processes were considered. The raw materials cost \$USD 7.2 to prepare a piece of sorbent in the lab, and the price of LiCl accounts for 86% of the total price, because the Sigma-Aldrich or Aladdin (Chemical supplier) ACS reagent chemical was used in the lab-scale preparation (170 \$/kg). However, if considering the industrialized mass production potential of LiCl, the price can be reduced by approximately 95% (8 \$/kg). The industry standard chemicals can still meet the demand regarding the purity level for our applications. Furthermore, the matrix activated carbon fiber, as one kind of specially treated carbon material, can be produced by being heated from recycled organic biomass (coconut, pistachio shell, saw dust, etc.), which is cheap and sustainable. Compared to the lab-scale preparation, purchasing larger amounts (on a hundred-kilogram scale) of raw materials can further reduce the price to \$USD 1.0. The results demonstrated the commercial potential and the sustainability of the sorbent.

The costs of the preparation process include the amortization expenses and the cost of labor-hour. It takes only \$0.65 to prepare a piece of sorbent (625 cm²), because the preparation of the sorbents only requires mild temperature drying and stirring dissolution, not involving expensive equipment, very high pressure/temperature conditions, and harmful processes to the human body or the environment. Overall, the cheap and industrially mass-produced raw materials and the simple production processes makes the sorbent highly suitable for mass production and show great market potential.

To sum up, the price of 8 piece of sorbents with the required surface area that was used in the water harvester sums to \$8.65 for mass production and \$58.25 for lab-scale fabrication. The total capital investment includes one water harvesting device and eight pieces of sorbents, which is \$249.45 for the lab prototype and \$107.65 for the mass-produced device.

Operating costs

The operating costs are estimated based on the field test data. With such sorption and desorption conditions, the water harvester produced 311.69 g freshwater and consumed 0.695 kWh electricity (i.e., ~2.2 kWh/L), which is competitive compared to the active dew water collection system (~6 kWh/L in semi-arid/arid regions and ~ 1 kWh/L in humid climates)^{6,7}. Based on this demonstration, the costs of 1-liter drinking water generation are ~\$ 0.19 according to the local average electricity costs.

Although the costs of operation/energy consumption highly relate to the working conditions, energy supply methods (grid, PV, or the breeze electricity), and the scale of the water harvester, it still shows a potential advantage over other complex water harvesters^{8,9}.

Changes made:

We added the description of the stability evaluation on **Page 8 & 14** and related **SI**, highlighted in red. “Sorption-desorption cycles performed over 186 hours confirmed the stable equilibrium and dynamic sorption performance without degradation and LiCl leakage (Fig. 2d and Supplementary Fig. 15).”

“Finally, the stability of the device was evaluated by conducting the 6-day water harvesting and batch-processed water production cycling experiments under various simulated climate conditions, further demonstrating the stability of the bulk sorbents and the strong environmental adaptability of the water harvester (Supplementary Fig. 28).”

We updated the experimental methods related to the stability evaluation in **Methods** part, highlighted in red.

“The sorption-desorption cyclic experiments were conducted to evaluate the stability of LiSHC. The dehydrated sample with an area of 3 x 3 cm² firstly absorbed water vapor at 15°C and 65% RH inside the constant climate chamber (KMF-115, Binder) for 12 hours. After that, the environment was changed to the desorption condition (80°C, 15% RH) for 6 hours. The sorption-desorption cycles were repeated ten times, which took over 10,000 mins. The weight change of the sample was recorded by the analytical balance. The stability and adaptability of the water harvester were evaluated by the 6-day water collection experiments under three various simulated climatic conditions. Detailed information can be found in Supplementary Information.”

We added the information on techno-economic assessment on **Page 12-13** and related SI, highlighted in red.

“Overall, the lightweight sorbent Li-SHC produced with low-cost and sustainable raw materials, facile and easily scaled-up fabrication procedures show excellent water sorption performance, adaptability and stability, which allows fast deployment into practical arid SAWH applications

(Supplementary Information Section 5). Additionally, the portable and cheap water harvester can be easily disassembled, re-assembled, and carried by a person. Therefore, when the demand..."

We added a new part about the water production potential estimation on **Page 16** and related SI, highlighted in red. We also combined a new figure **Fig. 5** to include **Fig. R5** and **R6**.

"For a comprehensive assessment of the water production potential, we selected five typical climates across the globe, including arid, semi-arid, and humid climates, and obtained their daytime and nighttime average ambient temperature and RHs (Fig. 5b and Supplementary Information Section 9). Then, based on the collected performance data in field tests, the water production of each location was estimated, showing that it is significantly influenced by the ambient RH, as over 1000 mL day⁻¹ water production could be achieved in a relatively humid climate (e.g., Birmingham, UK) but the lower limit of water production (150 mL day⁻¹) was reached in the aridest month of the Sahara Desert. In addition, the global daily water production potential of the water harvester is also estimated according to the DubibibAstakhov (D-A) equation⁶¹, the global average annual RH, and the collected performance data of the proposed water harvester. The water production potential of the water harvester is conservatively estimated at over 350 mL day⁻¹ in the majority of locations, only except for the Tibetan Plateau, North Africa, etc. (Fig. 5c).

Our developed water harvester with facile operation strategy shows its impressive recordbreaking metrics, demonstrating near realizable high yield of a portable water production anytime and anywhere. Featured with low total capital investment of both the sorbent and device and low operation costs, the portable water harvester demonstrated its market potential, making it closer to the marketization and industrialization."

"Fig. 5 Evaluation of a water harvester and estimated water production. a An evaluation of the presented water harvester in terms of comprehensively evaluated metrics based on existing sorbents and corresponding devices. b Estimated daily water production in typical arid, humid, and semi-arid climates. The month with the lowest (highest) monthly average RH in the whole year is marked as a low (high) RH month. c Daily water production of the water harvester estimated by the yearly average RHs."

2. The thermal design for an AWH system is critical, there are many AWH devices that use fans as the mass transfer enhancement measures. Is this water harvester also applying such measures to boost mass or heat transfer? Besides, in Figure 2B, is there also a convective heat transfer (heat loss) between the heating plate and the ambient air? Are there some thermal insulations between the heating surface and ambient? If possible, please clarify it in the text.

Response to Q1: Is this water harvester also applying such measures to boost mass or heat transfer?

We appreciate your comment to make our manuscript clearer. In our study, we do not apply any fans or condensers to enhance the mass or heat transfer in the harvester. Instead, many efforts, such as the numerically simulations related to the temperature and velocity fields inside the harvester, or adding the insulation panel, were taken to enhance temperature and concentration gradients to realize controlled natural air flow, leading to increased temperature difference between the sorbent and the condensation cover, which ensures the enhanced performance of the water harvester as demonstrated by the simulation. We compared system setups with two references that report complete water harvesters in details (ACS Cent Sci. 2019;5(10):1699-706 and Energy Environ. Sci., 2021,14, 5979-5994), as shown in Table R4.

Table R4. Comparison of the additional components in the water harvesters

	Nikita Hanikel et. al. ⁸	Xu et. al. ¹⁰	This work
Heat source	Electric heating strips	Solar thermal	Electric heating plate
Convection enhancement method	One large air fan & eight small fans	One air fan	/
Condensation enhancement method	A compressor-based condenser powered with four 12 V deep-cycle Batteries.	/	/
Others	/	Rotation by hands	/

For the first reference, one large and eight small fans were used to enhance the mass transfer inside the MOF-layer channels. A bulky compressor-based condenser was connected to the MOF-layer exchangers to decrease the condensation temperature and collect the desorbed water vapor as much as possible. These measures increased the amount of water collected, but also undoubtedly made the

harvester bulky and not portable. Similarly, in the second paper, a fan was also used to generate forced convection air to accelerate both the water sorption and water collection. Besides, the rotation of the device relies on the manual operation and may be further realized to quasi autonomous operation using additional electrical and mechanical components. These methods, including internal forced convection to increase heat and mass transfer and external forced convection to enhance heat dissipation and to reduce the condensation temperature undoubtedly further contribute to a higher final water collection, but at the expense of portability and compactness.

The presented conceptual prototype achieving water yield on a hundred-gram scale could serve as an inspiration for future research work related to batch-operated practical AWH systems. Currently, the desorption heat is provided by joule heating. The main advantage of proposed desorption mechanism based on electrical heating is to obviate the intermittence and periodicity of solar thermal heating to desorb and collect the adsorbed water at night as much as possible. Therefore, the thin-film solar cells that can be carried by a person could be used for portable water harvesting, and the silicon solar cells could power the scaled-up water harvester in areas without an electrical grid. New research avenues related to direct solar utilization in the form of the PV/T panels or individual solar absorbers in conjunction with thermal storage should be investigated in detail, which could improve the overall system energy efficiency. More importantly, storing solar energy in the form of electricity and thermal heat could expand the water release process into the night, realizing the all-day atmospheric water harvesting. Finally, further advancements of tailor-made sorbents, where new material with exceptional physical and chemical properties are combined into composite sorbents could pave the road for next-generation sorbents. For instance, joule-heating electrical components could serve as a nano/micro porous matrix, also possessing high thermal conductivity, thus further reducing losses by introducing electrical heat utilization.

Changes made:

As the editor suggested, we have incorporated all the information above and added a discussion on the limitations and potential developments of our proposed water harvester on **Page 17** and related **SI**, highlighted in red.

“The presented conceptual prototype achieving water yield on a hundred-gram scale could serve as an inspiration for future research work related to batch-operated practical AWH systems. Currently, the desorption heat is provided by joule heating. The main advantage of the proposed desorption mechanism based on electrical heating is to obviate the

intermittence and periodicity of solar thermal heating to desorb and collect the adsorbed water during the whole day as much as possible.

...

Furthermore, new research avenues related to direct solar utilization in the form of PV/T panels or individual solar absorbers in conjunction with thermal storage should be investigated in detail, which could improve the overall system energy efficiency. More importantly, storing solar energy in the form of electricity (e.g., batteries) and thermal heat (e.g., phase change materials) could expand the water release process into the night, realizing all-day atmospheric water harvesting. Finally, further advancements in tailor-made sorbents, where new materials with exceptional physical and chemical properties are combined into composite sorbents could pave the road for next-generation sorbents. For instance, joule-heating electrical components could serve as a nano/micro porous matrix, also possessing high thermal conductivity, thus further reducing losses by introducing electrical heat utilization.”

Response to Q2: In Figure 2B, is there also a convective heat transfer (heat loss) between the heating plate and the ambient air? Are there some thermal insulations between the heating surface and ambient?):

Thank you for pointing this out. Yes, there is one layer of thermal insulation between the heating plate and the ambient air to reduce the heat losses. The thickness of this insulation layer was ~15 mm, which consists of one layer of aluminum foil (0.5 mm) and a layer of Nitrile Butadiene Rubber (NBR)/Polyvinyl Chloride (PVC). Figure 2B has been modified to show the insulation layer clearly, as shown in Fig. R7.

Fig. R7 Energy balance diagram of the SAWH device during the water release-condensation process

Besides, according to the suggestions provided by the editor, we added the photos of the water harvester that works in the lab and show the demonstration of the PV-driven water harvester, as shown in Fig. R8.

Fig. R8 Photos of the water harvester. **a** Water harvester that is working in the lab. **b** Demonstration of the PV-driven water harvester.

Changes made:

We added the pictures of the working water harvester on **Page 10, Fig. 3b** and updated the details on **Page 10, Fig. 3c**, highlighted in red.

“Fig. **b** Photo of the water harvester and the data collector. **c** Energy balance of the SAWH device during the water release-condensation process.”

We added the information to outline more clearly which data were collected in a lab environment and which data were collected in a field test. The details information about the field tests were added to **Method** part.

Page 12: “The parameter optimization experiments were conducted in the real semi-arid region (Lanzhou, China, 36.017° N, 103.784° E). Details information can be found in Supplementary Fig. 23-26.”

Page 13-14: “The daytime RH in Lanzhou, China was recorded as extremely low values ~15% RH, while it increased to ~65% RH during the nighttime as the air temperature dropped to 15 °C (Fig. 4b). Eight pieces of sorbents were exposed to the ambient and absorbed water vapor during the night.”

Method part: “Field test in the semi-arid climate The field tests of the water harvester were conducted in a real semi-arid environment, which is located in Xiagouya Mountain, Lanzhou, China (36.017° N, 103.784° E) in September 2021. The indoor and outdoor temperature and RH were recorded by two temperature and RH sensors (COS-03, Renke) with an accuracy of

0.15°C and 1.5% RH. The parameter optimization procedure before the batch-process experiments was conducted within two experimental days. The dry weights of sorbents were between 37.90-39.65 g. All sorbents were exposed to the air for 12 hours from 20:00 to 08:00 the next day. After that, the weight of the sorbent was recorded by a balance (0.01 g), and the water uptake of the sorbents were calculated (1.83-2.19 g g⁻¹).

...

The number of cycles can be determined by the daily water demand and the allowed operating time, and the maximum recommended number of cycles is eight.”

3. The 8 times water release cycle was taken to pursue the maximum daily water yield. Is it possible or necessary to increase the number of cycles further? If there is only one water release cycle in limited duration, how about the water productivity of the portable device?

Response:

Thank you for your particular comment. The periodic replacement of sorbent (batch-process mode) leads to enhanced water yield, achieving over 300-gram daily water production. The reason why we applied this operation mode is that we noticed the gaps between slow sorption and fast desorption. The peak derivative sorption mass change is ~0.5 g/g/h, only 1/10 compared with the results of the desorption condition (~5 g/g/h). Therefore, the batch-process mode essentially maintains the desorption rate at the higher rate and closes the gap of the sorption/desorption rates as much as possible.

Regarding the selection of cycle times, the desorption duration was initially decided as ~1 hour (~1/10 of the sorption duration) according to the sorption/desorption rate ratio. Further, the pre-experiments that were shown in Supplementary Figure 25-27 were performed to verify our selection. In the practical cyclic experiment, taking into account the time lag of vapor condensation and the frequency of material replacement, each cycle takes about 1.35 hours, and 8 cycles take around 11 hours. After that, all sorbents were removed from the device and deployed for the overnight sorption. The number of cycles can be determined by the daily water demand and the allowed operating time, and the maximum recommended number of cycles is eight.

If limiting to one sorption-desorption cycle a day (no sorbent replacement), the water productions of 29.2 to 63.7 mL could still be realized when using different desorption time (1-3 hours) and number of

layers (1 to 2), as shown in Supplementary Figure 25. In this case, the calculated water productivity per gram of sorbent was 0.77 to 1.68 L/kg/day, illustrating the competitiveness of our device and sorbent material in comparison to other water harvesters that usually have a lower than 1.0 L/kg/day water yield, as shown in Supplementary Figure 29.

Changes made:

We updated the explanation about the cycle number in **Method** part.

“Taking the time lag of vapor condensation and the frequency of material replacement into account, each desorption duration was ~1.35 hours, and 8 cycles took around 11 hours. The desorption time was from ~08:00 to ~19:30. Note that the fourth cycle was longer than the others. This rest time is in line with the special situation, such as an emergency water supply, that is operated by a single person, indicating that each AHW cycle could be decoupled and paused. The number of cycles can be determined by the daily water demand and the allowed operating time, and the maximum recommended number of cycles is eight.”

We also updated the details information about the performance of the water harvester that works in the one-cycle mode in **SI: Supplementary Figure 25-27 and 29**.

Reviewer #2:

The manuscript entitled “Exceptional Water Production Yield Enabled by Batch-process Portable Water Harvester in Semi-arid Climate” by Shan et al. shows the development of a portable and modularized sorption-based water harvester with ultra-high-water release in one water harvesting cycle under semi-arid climate. Moreover, Wang and his collaborators designed and optimized the portable water harvester by combining simulations and experimental methods. The presented article contains both basic experimental and application simulation tests, which are vital to support the claim with reliable data and in turn pave the way towards the application of this material in the field of water production in semi-arid regions.

In general, the work is accurate and well presented as a Nature Communications article should be, moreover the presented results are certainly of interest to readers of this journal. However, some issues are supposed to be fully addressed to further improve the quality of this presented work.

Response:

We really appreciate the reviewer for finding the value of our work in the AWH applications in real semi-arid regions. We addressed all your valuable comments in details and provided the following response, which really helped us to improve the quality of this work.

1. In the abstract, “with ultra-high-salt-content sorbents up to 90 wt%” is mentioned, is this something that must be emphasized in the abstract? I think this should not be the focus and innovation of this work, so I suggest deleting it in the abstract.

Response:

Thank you for your suggestion. We re-wrote the abstract, which now highlights the performance and advantages of the prepared sorbent Li-SHC and water harvester.

Changes made:

We modified the ***abstract*** part, highlighted in red, as shown below.

“Sorption-based atmospheric water harvesting has the potential to realize water production anytime, anywhere to alleviate global water scarcity. However, virtually none of the research studies reached a hundred-gram water yield although state-of-the-art sorbents have been used. Here, we report a portable and modularized water harvester with scalable, low-cost,

and lightweight LiCl-based hygroscopic composite (Li-SHC) sorbents. Li-SHC achieved exceptional water uptake capacity of 1.18, 1.79, and 2.93 g g⁻¹ at 15%, 30%, and 60% RH, respectively. Considering the large mismatch between water capture and release rates, a rationally designed batch alternating mode is proposed to pursue maximum water yield in a single diurnal cycle. The portable water harvester shows a record water yield of 312 mL day⁻¹, 55.66 g_{water} L_{device}⁻¹ day⁻¹, and 97.40 g_{water} kg_{device}⁻¹ day⁻¹ in the semi-arid climate with the extremely low RH of ~15%, demonstrating the adaptability and possibility of achieving large-scale and reliable water production in real scenarios.”

2. Obviously, the material preparation in this study is very simple and does not seem to be innovative. The authors need to refine the unique advantages and innovations of this material in the manuscript.

Response:

Thank you for your suggestion. We highlighted the three unique advantages of the sorbent material: 1. High water uptake performance in a wide range of RH, especially in the low RHs; 2. Low-cost and ecofriendly raw materials; 3. Simple and easy scaled-up fabrication procedures. Thus, the excellent performance and scalability of the Li-SHC sorbent shows a particular potential for the mass production and practical field application.

High water uptake performance in a wide range of RH

The Li-SHC sorbent has shown the excellent water uptake performance in both lab environment chamber and field tests (Fig. 2b and 2e in the main text). In brief, benefitting from the record LiCl salt content 90 wt% and the strong hygroscopicity of LiCl salt, the sorption capacity of the sorbent Li-SHC reached over 846% of its initial weight at 90% RH within 12-hour sorption duration. Especially in the arid climate (15% RH), the sorbent Li-SHC shows the water uptake of 1.18 g/g, which is ~5 times higher water uptake than MOFs (MOF-303 or MOF-801) in similar real-world climate conditions. Some state-of-the-art AWH sorbents were selected for comparison at typical arid area climates (low RH, 30%) as shown in Fig. 2c. Overall, the sorbent Li-SHC shows strong environmental adaptability in various climates.

Low-cost and eco-friendly raw materials

Table R5 Costs of raw materials of the sorbent (unit: \$USD)

Raw materials	Price of one prototype	Price in case of mass production
Lithium chloride	6.4	0.3
Active carbon fiber felt	0.5	0.4
PTFE porous membrane	0.4	0.3
Total	7.2	1.0

The raw materials for preparing Li-SHC sorbent include LiCl, active carbon fiber felt, and PTFE porous membranes, which are all industrialized commercial products, and no complex and expensive chemicals are involved. We calculated the costs of raw materials of the sorbent, as listed in Table R5. The raw materials cost \$USD 7.2 to prepare a piece of sorbent in the lab, and the price can be reduced to only \$USD 1.0. Note that the price of LiCl accounts for 86% of the total price due to the ACS reagent chemicals was used in the lab-scale preparation. The price can be reduced by ~95% if using the industrialized mass produced LiCl. In addition to LiCl, the matrix activated carbon fiber, as one kind of specially treated carbon materials, can be produced by being heated from recycled organic biomass (coconut, pistachio shell, saw dust, etc.), which is cheap and sustainable. Overall, the sorbent Li-SHC shows its sustainability and commercial potential.

Simple, easily scaled-up fabrication procedures

Table R6 Costs of preparation processes of the sorbent (unit: \$USD)

Preparation process	Depreciation expense of the equipment	Specific energy consumption
Drying	0.04	0.42
Stirring	0.03	0.16
Total	0.65	

The preparation of the sorbents only requires mild temperature drying and string dissolution, not involving expensive equipment, very high pressure/temperature conditions, and harmful process to the human body or the environment. This makes the sorbent highly suitable for mass production and great market potential. Besides, the bulk and shape-adjustable sorbents can be easily carried and alternated during the batch-process field tests. Therefore, taking into account the amortization expenses and the cost of labor-hour, it takes only \$0.65 to prepare a piece of sorbent with the area of 25x25 cm². Overall, it demonstrated that the sorbent Li-SHC shows the satisfied scalability and adjustability, and its total price is attractive in comparison to other sorbents, like hydrogels or MOF sorbents.

Changes made:

We added the summary of the advantages of the sorbent on **Page 7 & 12** and related *SI*, highlighted in red. Besides, **Fig. 2c** and **Fig. 2d** were updated to show the excellent water uptake capacity and stability of the sorbent.

Page 7: “In the arid climates (30% RH), Li-SHC also shows satisfactory water uptake performance in comparison to other reported AWH sorbents (Fig. 2c and Supplementary Fig. 13).”

Page 12: “Overall, the lightweight sorbent Li-SHC produced with low-cost and sustainable raw materials, facile and easily scaled-up fabrication procedures show excellent water sorption performance, adaptability and stability, which allows fast deployment into practical arid SAWH applications (Supplementary Information Section 5). Additionally, the portable and cheap water harvester can be easily disassembled, re-assembled, and carried by a person. Therefore, when...”

Fig. 2: “**c** Comparison of vapor sorption capacity with the state-of-the-art sorbent materials at 30% RH. **d** Cycling performance of Li-SHC. The sorption and desorption are performed at 65% RH, 15 °C and 15% RH, 80 °C, respectively.”

3. In Line 66, “its production costs” should be changed to “their production costs”.

Response:

We are sorry for this language mistake. The related text has been revised.

4. In Line 124, the authors claim that “Since the porous.....water sorption”. I believe it would be better to provide more references in order to support the conclusion by previously published articles in that field.

Response:

Thank you for your suggestion. The developed sorbent Li-SHC can be regarded as a kind of salt-embedded composite sorbents, which is also called the “composite salt in porous matrix”¹¹. As the references reported^{10, 11, 12, 13}, the sorption characteristics of Li-SHC are mainly based on the

confined hygroscopic salts characteristics, which is the lithium chloride (LiCl) in this work. Generally, the vapor adsorption on the matrix surface is only responsible for 2-5% of total sorption after the salt loading, and this value is even lower for the sorbents with high salt content of 90 wt% in this work, because most sorption capacity of the matrix itself is lost due to the occupation of the matrix sites. As the result, the water sorption of the porous matrix can be negligible, and the sorption capacity of Li-SHC sorbents is mainly determined by the content of LiCl salt. The related references have been added in the revised manuscript.

Based on this, we further theoretically calculated the equilibrium water uptake and the sorption capacity in each step of the sorbent Li-SHC-90 according to the sorption capacity of LiCl and the salt content. Specifically for each sorption process, the sorbent Li-SHC-90 shows three-step sorption mechanisms during the sorption process (Fig. 2a in the main text).

The first step is the chemisorption of LiCl when the anhydrous LiCl crystals inside the composite sorbent capture water molecules and form hydrous salt ($\text{LiCl}\cdot n\text{H}_2\text{O}$, $n=1,2,3,5$). The value n is determined by the adsorption temperature. The phase diagram of LiCl and H_2O describe the crystallization lines of hydrous salts at different temperature. The crystallization line is described by the following equations, which is fitted according to the tested properties of the LiCl solution:

$$\theta = A \xi \frac{T}{T_c}$$

where ξ is the mass fraction of the salt solution, T_c is the critical temperature of water. The parameters A for different temperature and LiCl fraction ranges are included in Table R7.

Table R7 Parameters for the LiCl – H_2O crystallization line and phase diagram

Temperature range (°C)	LiCl fraction range	Crystal form	A	A	A
-75.5 to -68.2	0.253 to 0.287	$\text{LiCl}\cdot 5\text{H}_2\text{O}$	0.422088	-0.09041	-2.93635
-68.2 to -19.9	0.287 to 0.369	$\text{LiCl}\cdot 3\text{H}_2\text{O}$	-0.00534	2.01589	-3.11459
-19.9 to 19.1	0.369 to 0.452	$\text{LiCl}\cdot 2\text{H}_2\text{O}$	-0.56036	4.72308	-5.81105
19.1 to 93.8	0.452 to 0.558	$\text{LiCl}\cdot \text{H}_2\text{O}$	-0.31522	2.88248	-2.62433
over 93.8	over 0.558	LiCl	-1.31231	6.17767	-5.03479

This step contributes 0.42-0.85 grams of total water uptake and depends on the temperature. Typically for the sorbent, such as Li-SHC-90 operating under the ambient temperature (20-45°C), the monohydrate LiCl·H₂O is formed and contributes 0.38 g/g water uptake in the case of the salt content of 90 wt%.

Subsequently, the monohydrate salt crystals deliquesce and dissolve in the adsorbed water. The concentrated solution further absorbs water vapor and becomes dilute until the vapor pressure of the solution P is in equilibrium with the external vapor pressure in air (i.e., relative humidity). The equilibrium mass fraction ξ of the LiCl solution relates to the overall water uptake w , given by:

$$w = \frac{1 - \xi}{\xi}$$

The equilibrium mass fraction was determined by the ambient temperature and relative humidity, which is followed by the following equations:

$$P = \frac{C A B \theta}{P} \left(A - 2 \left(1 - \frac{\xi}{\pi} \right) \right) \left(B - 1 \left(\frac{\xi}{\pi} \right) \right) \left(C - 1 - 1 \frac{\xi}{\pi} \right) \pi e^{-\frac{1}{\pi}}$$

The parameters of π in above equations were listed in Table R8.

Table R8 Parameters for the vapor pressure equation

π_0	π_1	π_2	π_3	π_4	π_5	π_6	π_7	π_8	π_9
0.28	4.30	0.60	0.21	5.10	0.49	0.362	-4.75	-0.40	0.03

Finally, the sorption capacity of the sorbent with the salt content of x can be theoretically calculated as the following equation:

$$w = x w$$

Fig. R9 a Measured water uptake isotherm of Li-SHC-90, the matrix and theoretical isotherm of LiCl. (unit: $g_{\text{water}}/g_{\text{sorbent}}$). **b** Water uptake isotherm of pure porous matrix (unit: $g_{\text{water}}/g_{\text{matrix}}$)

Overall, the calculated theoretical isotherm and the experimental isotherm are shown in Fig. R8a, and the water vapor isotherm of the pure porous matrix was tested as shown in Fig. R9a and Fig. R9b. The results show essentially same isotherms of Li-SHC and LiCl, indicating LiCl is mainly responsible for the sorption capacity. The LiCl salt chemisorption and the salt solution absorption stage (salt solution zone) accounts for over 85% of total water uptake (when $RH > 60\%$), highly contributing to the water uptake performance. The water uptake of the pure porous matrix is relatively low $0.22 \text{ g}/\text{g}_{\text{matrix}}$ due to the micropore filling ($< 2 \text{ nm}$), and the water uptake increases with the increase of RH owing to the multilayer adsorption and capillary condensation in mesopores ($> 4 \text{ nm}$). Considering the high salt content (90 wt%) of the used composite sorbent, water uptake at 60% RH only attributes to $0.05 \text{ g}/\text{g}_{\text{sorbent}}$ (Fig. R9a), which could be negligible in comparison to the total water uptake (around $3 \text{ g}/\text{g}$ at this RH).

Changes made:

We added the explanation about the water uptake contribution on **Page 5 & 6** and related **SI**, highlighted in red.

“The abundant micropores of carbon fibers provide ultra-high specific surface areas (Supplementary Fig. 1), and their physical entanglement and numerous channels enhance the hydrophilicity (Supplementary Fig. 2) and capillary force of the felt (Supplementary Fig. 3). The water uptake isotherm of the pure porous matrix (Supplementary Fig. 4) demonstrates the micropore filling at the low RH ($< 30\%$) and an increased water uptake at higher RH owing to the multilayer adsorption and capillary condensation in mesoporous, but the relatively low

water adsorption capacity (< 0.1 g g⁻¹ below 30% RH) can hardly be suitable for water harvesting in arid climates. Therefore, hygroscopic nanoscale LiCl salts were uniformly loaded to the porous matrix by vacuum impregnation and mild heating methods to promote the water harvesting capacity in a wide working range of RHs. Meanwhile, the active carbon fiber felt serves as the matrix to support and disperse LiCl to alleviate the salt agglomeration and salt solution leakage. After LiCl loading, the totally dried sorbent becomes less flexible and shows a higher mechanical strength than the pure matrix. However, the flexibility of the sorbent can be recovered after sorption (Supplementary Fig. 5).

...

To reveal the sorption mechanism of Li-SHC, the phase diagram and the theoretical isotherm of pure LiCl were calculated (Supplementary Information Section 3) and presented in Supplementary Fig. 12. Obviously, the isotherm of Li-SHC shows highly similar isotherm of pure LiCl, which indicates that the equilibrium water uptake of Li-SHC is mainly determined by the sorption capacity of LiCl salt. Or to say, the porous matrix has a negligible contribution to the equilibrium water uptake. The water uptake of Li-SHC shows the multi-step mechanism as demonstrated in Fig. 2a: 1. Chemisorption of anhydrous LiCl below the deliquescence RH (11% RH at 30 °C); 2. Deliquescence of hydrous LiCl (LiCl·H₂O) and the saturated solution forms 3. Absorption of water vapor in saturated/concentrated LiCl solution. Note that over 85% of total water uptake was attributed by the adsorption of water vapor in LiCl solution when sorption at RH > 60%, thus the third step highly contributes to the overall water uptake capacity.”

5. In Figure 1E, how to calculate normalized water uptake?

Response:

The normalized water uptake was normalized to 1.0 for the equilibrium water uptake at each sorption condition (Table R9). It is calculated by the following equation:

$$w = \frac{w_t}{w_{t, 30^\circ\text{C}, 60\% \text{ RH}}}$$

Table R9 Equilibrium water uptake for samples with various salt content at 30 °C and 60% RH

Sample	Equilibrium water uptake at 30 °C, 60% RH
Li-SHC-40	1.47 g/g
Li-SHC-70	2.31 g/g

Li-SHC-80	2.64 g/g
Li-SHC-90	2.93 g/g
Li-SHC-95	3.10 g/g

Note that the equilibrium water uptake of four samples (Li-SHC-40, 70, 80, 90) was directly recorded at 12 hours, as shown in Fig. 1e. Specially, Li-SHC-95 cannot reach equilibrium at 12 hours due to the slow sorption dynamics. The equilibrium water uptake of Li-SHC-95 was directly acquired from the isotherm and is different from this value at 12 hours.

Changes made:

We deleted this figure because we used a dynamic sorption model to evaluate the dynamic characteristics of the sorbent. The information was updated on **Page 6**.

“The linear drive force (LDF) model was used to quantitatively evaluate their dynamic sorption characteristics (Supplementary Information Section 2). The sorption rate coefficients k_{LDF} indicates a strong positive correlation between the salt content and the sorption rate (Supplementary Fig. 10). This variation of dynamic sorption characteristic owes to the mass transfer enhancement effect, provided by the matrix, as it leverages the effect of additional pores and channels for increased sorption interfacial area. However, with the increased salt content, LiCl crystals form not only on the fiber pores but also on the surface and between the gaps of fibers (SEM images see Supplementary Fig. 11), which inevitably reduced the vapor transport channels and the sorption dynamics. Note that although reducing the salt loading can increase the sorption dynamics, the absolute amount of water uptake decreases simultaneously. The desired salt content in practical water harvesting applications can be optimized by matching the duration of a sorption cycle with the required equilibrium duration, indicating no waste of both the sorbent material and the operating time.”

6. Does the super-high salt content make the material brittle? How about the mechanical properties of the prepared material? How about the cycling performance? Does the salt content of the material decrease after repeated use? Generally, LiCl impregnated with materials is easy to lose. Is LiCl in the sorbent prepared in this paper easy to lose? How is LiCl preserved and stable and long-lasting in the material?

Response to Q1: Does the super-high salt content make the material brittle? How about the mechanical properties of the prepared material?

Thank you for the comment. The mechanical properties of the pure porous matrix, the dried as-prepared sorbent, and the sorbent that captured water vapor at 70% RH were evaluated by a dynamic mechanical analyzer with a loading rate of 20 mm/min.

Fig. R10 a Stress-strain curve of the pure porous matrix, the absorbed sorbent at 70% RH, and the dried as-prepared sorbent. Photos of the states of the b pure porous matrix, c dried matrix, and d absorbed sorbent when they are bent and then released.

The stress-strain curve is shown in Fig. R10a, indicating that the pure porous matrix shows excellent flexibility but relatively low breaking strength (0.095 MPa), indicating an elastic increase of the load before reaching the maximum stress. After that, the load decreased slowly with the displacement, which may result from the fiber-bridging and sliding after debonding and pulling out of carbon fibers¹⁴. Besides, as the photograph shows (Fig. R10b-d), the matrix can be easily bent and restored to its original state, demonstrating the flexibility of the pure matrix. Besides, a 1.3 kg box could be easily lifted by a 2 cm wide carbon fiber matrix without damage.

After the LiCl salt impregnation, the dried as-prepared sorbent that loaded ~90 wt% salt shows a high breaking strength (0.723 MPa), but demonstrates non-stretchable behavior, indicating a brittle failure mode. LiCl particles may act as an interface binder to transfer the load to the whole composite material¹⁵. The totally dried sorbent can be easily broken (Fig. R10c). However, after sorption at 70% RH, the sorbent becomes softer, and its flexibility is partly recovered (Fig. R10d) because of the formation and filling of the high concentration LiCl solution, showing a pseudoplastic behavior and an increased toughness^{14,16}.

Changes made:

We added the description about the mechanical properties of the sorbent on **Page 6**, and more details are added in **SI: Supplementary Figure 5**.

“After LiCl loading, the totally dried sorbent becomes less flexible and shows a higher mechanical strength than the pure matrix. However, the flexibility of the sorbent can be recovered after sorption (Supplementary Fig. 5).”

Response to Q2: How about the cycling performance? Does the salt content of the material decrease after repeated use? Generally, LiCl impregnated with materials is easy to lose. Is LiCl in the sorbent prepared in this paper easy to lose? How is LiCl preserved and stable and long-lasting in the material?

The stability of the material and device is indeed an important issue in a long-term application, and we comprehensively did the stability evaluations of both the device and the material.

Fig. R11 Stability evaluation of the sorbent. **a** 180-hour sorption-desorption cycling tests of the sorbent Li-SHC at 65% RH, 15 °C for sorption and 15% RH, 80 °C for desorption. **b** Photos of the sorbent Li-SHC after exposing to the high RH of 90% RH, showing the solution leakage was avoided by the membrane encapsulation.

Fig. R12 Stability evaluation of the device at various simulated climate conditions. **a** Photos of the produced freshwater on each day. **b** Collected freshwater amounts of each cycle and each day. **c** Overall water production during the cyclic tests. The insert shows the photos of the overall collected water on each day **d** Overall energy consumption during the cyclic tests. **e** Mass changes of each piece of the sorbent during the 6-day water capture-release cycling tests. **f** Temperature variation of the

heating plate, sorbent, chamber, and two measuring points on the condenser. The measuring points of the sorbent moved due to released vapor turbulence, resulting in the measured temperature fluctuations. To evaluate the stability of the sorbent, we performed a 180-hour water sorption-desorption cycling experiments using the sorbent with the area of 9 cm² inside the constant environmental chamber. The sorption and desorption condition are 65% RH, 15 °C and 15% RH, 80 °C, respectively. As shown in Fig. R11a, no obvious performance degradation was found during the sorption-desorption cycle. Note that the water uptake of the first sorption cycle is less than the remaining cycles as in the first active stage the sorbent absorbed water vapor from the total dry state.

In addition, the risk of the salt solution was tested at high RH (75% RH and 95% RH), showing that no solution leakage was observed even at RH as high as 95% (Fig. R11b). This is because the solution was totally encapsulated inside a waterproof but moisture permeable membrane with the average pore size of 10 μm. The humidity in the air can be permeable and adsorbed by the sorbent, but the generated salt solution cannot cross through the waterproof membrane to leak, which ensuring the stability and reliability of the sorbent Li-SHC. The membrane encapsulation can bring other advantages, such as avoiding the metal corrosion and preventing water quality deterioration.

To evaluate the stability of the device, we conducted a 6-day water harvesting and batch-processed water production cycling experiments under three typical climate conditions in our field test location, Lanzhou, China. The three nighttime sorption conditions are the typical climate in the spring/autumn season (15°C, 65% RH), the typical climate in the winter season (5°C, 50% RH), and the typical climate in the summer season (20°C, 75% RH). Experiments under each climate condition were conducted for 2 days, with a total experimental time of 6 days. As shown in Fig. R12a and Fig. R12b, the water production remained consistent in each climate condition, which are ca. 340 g at the spring condition, ca. 280 g at the winter condition, and ca. 425 g in the summer condition. The total water production over the 6 days is ~2115 g (Fig. R12c), showing the excellent water production performance and the adaptivity of the water harvester to different climates. As shown in Fig. R12d, the total energy consumption is 5.97 kWh, corresponding to the water production of ~355 g per kilowatt-hour.

In addition to the overall water production and energy consumption, the mass changes of the sorbents during the cycles are shown in Fig. R12e. It shows that the average water uptake of each sorbent is 2.41 g/g, 2.04 g/g, and 2.81 g/g for each climatic condition, respectively. More importantly, all sorbents have no obvious mass loss after these cycles, showing excellent stability throughout the cyclic tests with variable working conditions. Detailed information on the temperature variation of each component of the water harvester can be found in Fig. R12f.

Changes made:

We added the description of the stability evaluation on **Page 8 & 14** and related **SI**, highlighted in red.

Page 8: “Sorption-desorption cycles performed over 186 hours confirmed the stable equilibrium and dynamic sorption performance without degradation and LiCl leakage (Fig. 2d and Supplementary Fig. 15).”

Page 14: “Finally, the stability of the device was evaluated by conducting the 6-day water harvesting and batch-processed water production cycling experiments under various simulated climate conditions, further demonstrating the stability of the bulk sorbents and the strong environmental adaptability of the water harvester (Supplementary Fig. 28).”

We updated the experimental methods related to the stability evaluation in **Methods** part, highlighted in red.

“The sorption-desorption cyclic experiments were conducted to evaluate the stability of LiSHC. The dehydrated sample with an area of 3 x 3 cm² firstly absorbed water vapor at 15°C and 65% RH inside the constant climate chamber (KMF-115, Binder) for 12 hours. After that, the environment was changed to the desorption condition (80°C, 15% RH) for 6 hours. The sorption-desorption cycles were repeated ten times, which took over 10,000 mins. The weight change of the sample was recorded by the analytical balance. The stability and adaptability of the water harvester were evaluated by the 6-day water collection experiments under three various simulated climatic conditions. Detailed information can be found in Supplementary Information.”

7. Authors claim that “A water vapor breathable but waterproof membrane...”. How is the moisture permeability of PTFE membrane? Please supplement this data in Supplemental Information.

Response:

Thank you for your suggestion. In this study, a commercial hydrophobic but moisture permeable membrane with a thickness of 20 μm and porosity of 0.87 was selected for the prevention of salt solution leakage. The average pore size of the porous membrane is 3 μm. The measured lower limit of the air permeability is 127.4 mL/(cm²·min). We assumed that the water vapor molecules are uniformly carried by the air through the membrane. The concentration of water in the air at the temperature of our field test (15°C) is in range from 0 to 12.82 g/m³ (0-1.282 × 10⁻⁵ g/mL) in relative humidity range

0%-95%. Therefore, the lower limit of the water molecular permeability is 0-980 g/(m²·h), as shown in Fig. R13.

Fig. R13 Lower limit of the water vapor molecules permeability at different RH.

Theoretically, the mass transfer process relied on the vapor diffusion inside the membrane pores, which has been discussed extensively in the literature as a combined effect of Knudsen diffusion, molecular diffusion, and viscous (Poiseuille) flow^{17,18}. The dimensionless Knudsen number (Kn) can be a guideline to determine the domain mechanism, which is defined as the ratio of the mean free path of the gas l [m] to the pore size r [m], as follows:

$$\text{Kn} = \frac{l}{2r}$$

The mean free path of the gas (water vapor) is the average traveled distance of the molecules between two collisions, given by:

$$l = \frac{kT}{\sqrt{2}\pi P\sigma}$$

where k_B is the Boltzmann constant 1.38×10^{-23} [m² kg s⁻² K⁻¹], T the temperature [K], σ the collision diameter (0.2641 nm for water vapor), P the mean pressure in membrane pores [Pa]. The mean free path is estimated to be 0.133 μm in our study.

When the vapor diffusion through the membrane and the pore size is smaller than the water vapor mean free path ($\text{Kn} > 1$), the molecule-pore wall collisions dominate the mass transfer, which is called as Knudsen region. Oppositely, when pores are much bigger than the mean free path ($\text{Kn} < 0.01$), the molecule-pore wall collisions are negligible¹⁹. Here in our study, the estimated Kn number is 0.04, which means the mechanism of vapor transfer inside the membrane belongs to the transitional region with $0.01 < \text{Kn} < 1$ ²⁰.

Changes made:

We added the information about the data in *SI: Supplementary Figure 9*.

8. In Line 394, “2.37 g/g” needs to be bracketed.

Response:

Thank you for your comment. We modified it in the revised manuscript. Meanwhile, the conclusion part was modified to highlight the advantages and main conclusions.

Changes made:

We updated the *Conclusion* part:

“We demonstrated a realization of a high-performance and portable water harvester with scalable, low-cost, and lightweight sorbent Li-SHC, which was comprehensively developed from the perspective of materials, as well as advanced structure design and operation strategy. With the optimized LiCl content, Li-SHC shows the water uptake capacity of 1.18, 1.79, and 2.93 g g⁻¹ at 15%, 30%, and 60% RH, respectively. The bulk sorbent Li-SHC shows a high water release of 2.37 g g⁻¹ in a single water capture-release cycle under a simulated practical semi-arid environment. The water harvester was designed and optimized by combining simulations and experimental approaches, achieving portable and high-yield water harvester without complex enhancement components. Considering the mismatch of adsorption and desorption dynamics, an eight-cycle batch alternating mode was proposed. This prototype was tested in a semi-arid region with an extremely low RH of 15% and achieved an exceptional water production yield of 312 mL day⁻¹ and 1.09 g_{water} g_{sorbent}⁻¹ day⁻¹, which puts the daily water productivity of portable sorption-based water harvester on the hundred-grams scale. Remarkably, this portable device shows apparent superiorities referring to the weight and space of the whole device, whose values are 55.66 g_{water} L_{device}⁻¹ day⁻¹ and 97.40 g_{water} kg_{device}⁻¹ day⁻¹, respectively, demonstrating exceptional performance considering these metrics. We anticipate that this study could make a closer step to practical AWH to meet the daily personal water demand.”

9. In Figure 3B, why does the fourth cycle take longer than the others?

Response:

Yes, it took 40 min longer in the fourth, the so-called stand-by stage. The main purpose of this rest time is to demonstrate a special situation, such as emergency water supply that is operated by a single person. It indicates that each AHW cycle could be decoupled and paused. The number of cycles can be determined by the daily water demand and the allowed operating time, and the maximum recommended number of cycles is eight.

Changes made:

We updated the explanation about the cycle number on **Method** part.

“Taking the time lag of vapor condensation and the frequency of material replacement into account, each desorption duration was ~1.35 hours, and 8 cycles took around 11 hours. The desorption time was from ~08:00 to ~19:30. Note that the fourth cycle was longer than the others. This rest time is in line with the special situation, such as an emergency water supply, that is operated by a single person, indicating that each AHW cycle could be decoupled and paused. The number of cycles can be determined by the daily water demand and the allowed operating time, and the maximum recommended number of cycles is eight.”

10. The use of “first” in the article should be avoided as much as possible. Please see Lines 332 and 401.

Response:

Thank you for your suggestion. We modified the text from “which is the first portable hundred-gramscale water-yield sorption-based atmospheric water harvesting device” to “which puts the daily water productivity of portable sorption-based water harvester on the hundred-grams scale”.

Changes made:

We updated the **Conclusion** part, as shown in **Question 8**.

11. Figure 4B seems more appropriate to be placed in the Supplemental Information. Or the authors can find some supplementary interpretation of the figures 4A and 4B and combine them into new Figure 4.

Response:

Thank you for your suggestions. Figure 4B has been moved to the Supplementary Information. The figure in the main text has been modified to show a figure that is more related to the comprehensive comparison of the water harvester and the water production estimation.

Changes made:

We added a new part about the water production potential estimation on **Page 16** and related *SI*, highlighted in red. We also combined a new figure **Fig. 5** to include **Fig. R5** and **R6**.

“For a comprehensive assessment of the water production potential, we selected five typical climates across the globe, including arid, semi-arid, and humid climates, and obtained their daytime and nighttime average ambient temperature and RHs (Fig. 5b and Supplementary Information Section 9) making it closer to the marketization and industrialization.”

“Fig. 5 Evaluation of a water harvester and estimated water production. a An evaluation of the presented water harvester in terms of comprehensively evaluated metrics based on existing sorbents and corresponding devices. b Estimated daily water production in typical arid, humid, and semi-arid climates. The month with the lowest (highest) monthly average RH in the whole year is marked as a low (high) RH month. c Daily water production of the water harvester estimated by the yearly average RHs.”

12. Figures 1F, 1L, 3C, 3D and 3E report interesting quantitative data that are very useful to prove the final applicability of the developed material. Anyway, the bars representing the standard deviations should be included.

Response:

Thank you for your comment. In the revised manuscript, the error bars were calculated and added to the average values, while the standard deviations have been added to related figures.

Reviewer #3:

The manuscript extensively describes an investigation aimed at developing a device for water harvesting from the atmosphere in dry diurnal climates such as in deserts. The investigation effort goes above and beyond, from material preparation all the way through device testing in a real setting.

Response:

Thank you very much for your positive comments. We also thank the reviewer's comments that help us to improve the quality of our work.

The most noticeable outcomes of this research consist in the exceptionally high water working capacity and the record daily water yield (per gram of sorbent, per kilogram of device and per volume of device), which also constitute the two main claims of this research.

It is worth noticing that the results apparently stand out when compared with the following two publications:

- 1) Hanikel N, Prevot MS, Fathieh F, Kapustin EA, Lyu H, Wang H, et al. Rapid Cycling and Exceptional Yield in a Metal-Organic Framework Water Harvester. ACS Cent Sci. 2019;5(10):1699-706
- 2) Xu J, Li T, Yan T, Wu S, Wu M, Chao J, et al. Ultrahigh solar-driven atmospheric water production enabled by scalable rapid-cycling water harvester with vertically aligned nanocomposite sorbent. Energy & Environmental Science. 2021

The second reference above shares similarities with the manuscript in the salt used to impregnate the matrix (LiCl), in the research approach (also in this case the characterisation of the material was followed by the characterisation of the structure and final testing in a real environment) and in the performance achieved by the material (Figure S23).

Response:

We appreciate your valuable summary and comments. As you mentioned above, our study was focused on the synergistic development of water harvesters from materials synthesis to the advanced heat/mass transfer optimization and operation strategies. This is because we noticed many previous works in the AWH field primarily emphasized the novelty and performance of the composite material, especially the equilibrium water uptake of the sorbents (usually tested in the gram scale) instead of focusing on the design of the water harvester and the overall water production of a prototype.

To be more specific, two important aspects are ignored in most of the research. One is related to the dynamic sorption/desorption performance of bulk materials in real life tests, as we have demonstrated in the main text. The dynamic performance has a strong impact on the final water collection performance, especially in practical multicyclic operational mode when using bulk several-gram sorbents instead of milligram lab-tested sorbents. Another aspect is referring to the device-level

optimization, especially in terms of the natural temperature gradient and heat dissipation of the AWH unit, which has been limited to the development of the portable device. These two points resulted in a common and obvious consequence, which is that these AWH water harvesters lost the portability and show unsatisfied gram-scale daily water production that is far from meeting the personal demand.

As for the two articles that the Reviewer highlighted, the researchers are aware of the significance of the device-level design and the operation strategy to boost the final water production yield. In fact, these two works are few studies that involved device-level optimizations and/or field tests. These two papers are definitely very important papers, providing significant progress in the AWH research field, with our study taking a step further.

Fig. R14 Location of water harvester field tests
(Xiagouya Mountain, Lanzhou, China, 36.017° N, 103.784° E)

Table R10 Evaluation of daily water harvesting performance

	Nikita Hanikel et. al. ⁸	Xu et. al. ¹⁰	This work
Daily water production (mL)	405.3	22.8	311.7
Water yield per mass of sorbent (L/kg/day)	0.7	1.05	1.09
Relative humidity	10-75%	/	15-65%
Location	Mojave Desert, USA (Arid region)	Shanghai, China (Humid region)	Lanzhou, China (Semi-arid region)

In the first paper, Yaghi's group developed a water harvester that operated utilizing multiple cycles during the daytime. They provided the mass transfer optimization of MOF particles using the cartridge structure to disperse MOF particles into layers of several milligram. The practical atmospheric water harvesting tests were conducted in Mojave Desert (~10-75% RH) and showed the average water production of 405.3 g (579 g_{MOF}, 0.7 L/kg_{MOF}·day). We carefully analyzed the water production data during the three experimental days and found that the high water production yield was mainly contributed by the first desorption cycle, since the experiment started with an equilibrated MOF bed that absorbed water vapor at the nighttime high temperature (extremely 75% RH). However, the low humidity during the day and the limited heat and mass transfer of the packed MOF layers seriously affected the water production during the daytime, and only a small amount of water (<0.1 kg_{water}/kg_{MOF}) is produced during the daytime. In the second paper, Xu et al. developed a solar-driven multicycle water harvester equipped with vertically aligned nanocomposite sorbents, which showed a water collection of 22.8 g (21.8 g_{sorbent}, 1.05 L/kg_{sorbent}·day) in a typical semi-humid climate (Shanghai, China). Note that the real field test was absent in this study.

Overall, the first advantage of our device is that our water harvester shows the superiority with regards to the total water production (312 mL) and the water yield per mass of sorbent (1.09 L/kg·day) in the lowest diurnal RH of 15%. The water production data was real tested in the semi-arid region instead of in the lab or humid areas. Such low-RH and remote region enables the full potential of sorptionbased water harvesting technology. The detail evaluation is shown in Table R10, and the location of the field test is shown in Fig. R14.

Table R11 Evaluation of water harvesting performance of sorbents in various RHs

	Nikita Hanikel et. al. ⁸	Xu et. al. ¹⁰	This work	Improvement compared to MOF-303
Sorbent type	MOF-303	LiCl@rGO-SA	Li-SHC	
Water uptake at 15% RH	0.40 g/g	1.01 g/g	1.18 g/g	295%
Water uptake at 30% RH	0.49 g/g	1.52 g/g	1.79 g/g	365%
Water uptake at 60% RH	0.52 g/g	2.76 g/g	2.93 g/g	563%

As we emphasized above, both the high-performance but low-cost sorbent and the optimized device contributed to the overall high water production. This accentuates the following two advantages of our study, which are related to materials and advanced system design of presented device. In terms of the material performance, the developed Li-SHC composite sorbents showed the water uptake of

1.18, 1.79, and 2.93 g/g at 15%, 30%, and 60% RH (Table R11). Besides, compared to the packed bed of MOF crystals, the sorbent that is supported by the carbon-based matrix is expected to achieve better heat transfer dynamics, which contributes to the water release.

Table R12 Comparison of the additional components in the water harvesters

	Nikita Hanikel et. al. ⁸	Xu et. al. ¹⁰	This work
Heat source	Electric heating strips	Solar thermal	Electric heating plate
Convection enhancement method	One large air fan & eight small fans	One air fan	/
Condensation enhancement method	A compressor-based condenser powered with four 12 V deep-cycle Batteries.	/	/
Others	/	Manually rotated	/

In terms of the entire device, we numerically simulated temperature and velocity distributions inside the water harvester and optimized the structure to enhance vapor transportation and condensation without additional energy consumption. The enhanced temperature and vapor concentration gradients inside the chamber were the driving force for enhanced desorption and condensation, ensuring obviation of applying mechanical components, such as fans or condenser. This design simplification ensured the portability and stability of the water harvester (Fig. 3a-b). The detailed comparison of experimental setups used among mentioned references and this work (Table R12) shows that for the first research study one large fan and eight small fans were used to enhance the mass transfer inside the MOF-layer channels. An unwieldy compressor-based condenser was connected to the MOF-layer exchangers to decrease the condensation temperature and collect the desorbed water vapor as much as possible. These measures increased the amount of water collected, but also undoubtedly made the harvester bulky and not portable. Similarly, in the second paper, a fan also was used to generate forced convection air to accelerate both the water sorption and water collection. Besides, the rotation of the device relies on the manual operation and may be further realized to quasi autonomous operation using additional electrical and mechanical components. These methods, including internal forced convection to increase heat and mass transfer and external forced convection to enhance heat dissipation and to reduce the condensation temperature undoubtedly further contribute to a higher final water collection, but at the expense of portability and compactness.

The presented conceptual prototype achieving water yield on a hundred-gram scale could serve as an inspiration for future research work related to batch-operated practical AWH systems. Currently, the desorption heat is provided by joule heating. The main advantage of the proposed desorption mechanism based on electrical heating is to obviate the intermittence and periodicity of solar thermal heating to desorb and collect the adsorbed water during the whole day as much as possible. Considering the high correspondence between arid regions and solar-rich regions, the possibility of using solar PV systems to boost low-carbon water harvesters was estimated (Supplementary Information Section 10). By matching the energy demand of the above 8 water harvesting cycles and the performance of commercial PV modules under different weathers, we calculated the demand for solar panel areas, showing that solar panels with an area of 1.11-2.08 m² can meet the requirements of the water collection cycle shown above. Therefore, the thin-film solar cells that can be carried by a person could be used for portable water harvesting, and the silicon solar cells could power the scaled-up water harvester in areas without an electrical grid. Furthermore, new research avenues related to direct solar utilization in the form of PV/T panels or individual solar absorbers in conjunction with thermal storage should be investigated in detail, which could improve the overall system energy efficiency. More importantly, storing solar energy in the form of electricity (e.g., batteries) and thermal heat (e.g., phase change materials) could expand the water release process into the night, realizing all-day atmospheric water harvesting. Finally, further advancements in tailor-made sorbents, where new materials with exceptional physical and chemical properties are combined into composite sorbents could pave the road for next-generation sorbents. For instance, joule-heating electrical components could serve as a nano/micro porous matrix, also possessing high thermal conductivity, thus further reducing losses by introducing electrical heat utilization.

Changes made:

We highlighted that “*the water harvester was designed and optimized by combining simulations and experimental methods, achieving potable and high-yield water harvesting without complex enhancement components*” in the **Conclusion** part. Also, we added the additional information on **Page 15**.

“ultra-high 97.40 gram per kilogram of device ($\text{g}_{\text{water}} \text{kg}_{\text{device}}^{-1} \text{day}^{-1}$) and 55.66 gram per liter of device ($\text{g}_{\text{water}} \text{L}_{\text{device}}^{-1} \text{day}^{-1}$). These advantages can be attributed to the advanced thermal design, which enables our devices to avoid the use of complex heat and mass transfer systems, such as active condensers and fans.”

This is not surprising since both the carbon-supported material of this manuscript and the carbon-supported material of the second reference have essentially the same isotherm, which is the isotherm of the pure LiCl given that the composite material is impregnated with LiCl above 15% in weight. 15% is usually the boundary between a composite material with intermediate equilibrium properties between support and salt and a composite having the equilibrium properties of the salts, scaled down by the amount of salt impregnated.

LiCl salt has deliquescence at 11.3% Relative Humidity at 30°C that is exactly what Fig. 1F shows. The salt isotherm is not particularly sensitive to the temperature (nothing beyond the sensitivity of the relative humidity to temperature). In fact, increasing the temperature to 100°C, the deliquescence only moves to 9.9% Relative Humidity. However, entering the conditions of Fig. 3A (adsorption at about 50% relative humidity average; desorption at about 20% relative humidity average) in the isotherm of Fig. 1F shows that in the real tests, the material worked in the salt solution zone. Besides, in a location where the day/night relative humidity swing is 30%, much more than in the location of Xu et al. Energy & Environmental Science (2021), so likely leading to better results.

Working in the salt solution zone makes wondering about the stability of the material. This important information is not present in the manuscript. The 8 samples of Fig. 3B have been used only once. Information whether or not the material can be used multiple times should be given. This would require also a better specification of the micropore size, whether crystals are also in the pores and their influence (likely negligible given the high loading) on the overall behaviour of the material. This raises also a question whether the carbon fibre support is really needed.

Response:

We agree with your insightful summary of the sorption characteristics of LiCl-based sorbents. We would like to illustrate the effects of the hygroscopic salt (LiCl) and the matrix (ACFF) on the LiCl-based carbon-supported sorbent in three aspects. And then, the stability of both the sorbents and the device are evaluated by the cyclic water capture-release experiments.

The effect of the matrix on sorption characteristics

Owing to the high salt content of ~90 wt%, our sorbent Li-SHC exhibits highly similar water uptake **isotherm** as pure LiCl, and the porous matrix makes less effect on the **equilibrium** water uptake, which is consistent with the results of other sorbents with similar synthetic methods^{10, 11, 12, 13}. However, if

considering the **dynamic** sorption process, the matrix influences much of the sorption dynamics due to the nanoscale interaction between the surface and the vapor/salt solution transport channels provided by the matrix. When LiCl is loaded, it partially occupies the transport channels, resulting in a drop in sorption dynamics. In other words, the amount of salt loading (the ratio of salt and matrix) reflects the effect of the matrix on the sorption dynamics.

The linear driving force (LDF) model was used to quantitatively evaluate the sorption dynamics of the samples with different salt contents. LDF model assumes that the mass transfer resistances are lumped in a film inside the adsorbent particle^{21, 22}. The heat transfer is assumed to be lumped in a film at the outside surface of the adsorbent. The diffusion rate into the adsorbent is essentially proportional to the difference between the equilibrium state and the current adsorbed state. The mass transfer governing equation is:

$$\frac{dC}{dt} = k(C^* - C)$$

The boundary conditions and the equilibrium relationships are described as:

$$\begin{aligned} t = 0, C &= C_0 \\ t = \infty, C &= C^* \\ C &= C^* p, T \\ C &= C^* p, T \\ C^* &= C^* p, T \end{aligned}$$

C is the absorbate weight per unit weight of the absorbent at the time t . C_0 is the initial absorbate weight per unit weight, and C^* denotes the equilibrium water uptake of a sorbent with various salt content, which is summarized in the Table R13.

Table R13 Equilibrium water uptake for samples with various salt content at 30 °C and 60% RH. Note that the equilibrium water uptake of the four samples (Li-SHC-40, 70, 80, 90) was directly recorded at 12 hours, as shown in Fig. 1e. Specially, LiSHC-95 did not reach equilibrium at 12 hours due to the slow sorption dynamic. The equilibrium water uptake of Li-SHC-95 was different from this value at 12 hours.

Sample	Equilibrium water uptake at 30 °C, 60% RH
Li-SHC-40	1.47 g/g
Li-SHC-70	2.31 g/g
Li-SHC-80	2.64 g/g
Li-SHC-90	2.93 g/g
Li-SHC-95	3.10 g/g

In our cases, the temperature and pressure of the adsorbent are assumed to be uniform and constant due to the relatively slow sorption process. Hence, T , p , and C_0 . Besides, the initial water uptake is 0 because sorbents are put into an oven at the temperature of 120 °C for the dehydration process. Because of the uniformity of sorbents, the equilibrium sorption capacities of all samples can be identical. Hence, the LDF model can be expressed as:

$$C - C_0 = 1 - \exp(-kt)$$

where k is the sorption rate coefficient that can be used to evaluate the sorption kinetics of different samples.

In the above equation, the sorption rate coefficient k can be used to evaluate the sorption dynamics. We fitted the sorption dynamic curves of Li-SHC-40, 70, 80, 90 and obtained the rate coefficient, as shown in Table R14 and Fig. R15.

Table R14 Sorption rate coefficients for samples with different salt contents

Sample	Sorption rate coefficient k (s^{-1})
Li-SHC-40	2.07×10^{-3}
Li-SHC-70	6.86×10^{-4}
Li-SHC-80	4.50×10^{-4}
Li-SHC-90	2.40×10^{-4}
Li-SHC-95	1.07×10^{-4}

Fig. R15 Dynamic experimental sorption curves and LDF model sorption curves of samples with the salt content of **a** 40 wt%, **b** 70 wt%, **c** 80 wt%, **d** 90 wt%, **e** 95 wt%, and **f** the summary of the LDF coefficient.

The largest and smallest k obtained in the samples with highest and lowest salt content, respectively, indicates a strong positive correlation between salt content and sorption rate. Intuitively, as demonstrated in Fig. 1e, the equilibrium sorption duration increased from ~4 hours to over 12 hours with the salt content increased from 40 to 95 wt%. This variation of dynamic sorption characteristic owes to the mass transfer enhancement effect, provided by the matrix, as it leverages the effect of additional pores and channels for increased sorption interfacial area. However, with the increased salt content, lithium chloride crystals are formed not only on the fiber pores but also on the surface and between the gaps of fibers, which inevitably reduces the sorption dynamics. That is why we evaluated the equilibrium and dynamic sorption performance of sorbents with different salt contents and finally selected the sorbent with the salt content of 90 wt%, which shows the best tradeoff between the dynamic and equilibrium performance. Therefore, the active carbon fiber matrix forms the bulk sorbent (Li-SHC composite sorbent), which is easy to use in the AWH device. Also, it avoids the direct utilization of the liquid absorbent (LiCl solution). More importantly, the provided reaction surface and vapor transport channels of the composite solid sorbent can accelerate the sorption process, obtaining a faster water uptake dynamic.

The effect of LiCl on sorption characteristic

The equilibrium water uptake of the sorbent is controlled by the sorption characteristics of LiCl and its content. As the Reviewer mentioned, the sorbent operates in the salt solution zone, which can also be defined as a salt solution adsorption region (Fig. 2a). Specifically, the sorbent Li-SHC-90 shows three water sorption mechanisms during the sorption process (Fig. 2a), which are chemisorption, deliquescence, and salt solution adsorption. Based on this, we further theoretically calculated the equilibrium water uptake and the sorption capacity in each step of the sorbent Li-SHC-90 according to the sorption capacity of LiCl and the salt content.

The first step is a chemisorption of LiCl when the anhydrous LiCl crystals inside the composite sorbent capture water molecules and form hydrous salt ($\text{LiCl} \cdot n\text{H}_2\text{O}$, $n=1,2,3,5$). The value n is determined by the adsorption temperature. The phase diagram of LiCl and H_2O describe the crystallization lines of hydrous salts at different temperature. The crystallization line is described by the following equations, which is fitted according to the tested properties of LiCl solution:

$$\theta = A \xi$$

$$\theta \stackrel{\text{def}}{=} \frac{T}{T_c}$$

where ξ is the mass fraction of the salt solution, T_c is the critical temperature of water. The parameters A for different temperature and LiCl fraction ranges are included in Table R7.

Table R14 Parameters for the LiCl – H₂O crystallization line and phase diagram

Temperature range (°C)	LiCl fraction range	Crystal form	A	A	A
-75.5 to -68.2	0.253 to 0.287	LiCl·5H ₂ O	0.422088	-0.09041	-2.93635
-68.2 to -19.9	0.287 to 0.369	LiCl·3H ₂ O	-0.00534	2.01589	-3.11459
-19.9 to 19.1	0.369 to 0.452	LiCl·2H ₂ O	-0.56036	4.72308	-5.81105
19.1 to 93.8	0.452 to 0.558	LiCl·H ₂ O	-0.31522	2.88248	-2.62433
over 93.8	over 0.558	LiCl	-1.31231	6.17767	-5.03479

This step contributes 0.42-0.85 grams of total water uptake and depends on the temperature. Typically for the sorbent, such as Li-SHC-90 operating under the ambient temperature (20-45°C), the monohydrate LiCl·H₂O is formed and contributes 0.38 g/g water uptake in the case of the salt content of 90 wt%.

Subsequently, the monohydrate salt crystals deliquesce and dissolve in the adsorbed water. The concentrated solution further absorbs water vapor and becomes dilute until the vapor pressure of the solution P is in equilibrium with the external vapor pressure in air (i.e., relative humidity). The equilibrium mass fraction ξ of the LiCl solution relates to the overall water uptake w , given by:

$$w = \frac{1 - \xi}{\xi}$$

The equilibrium mass fraction was determined by the ambient temperature and relative humidity, which is followed by the following equations:

$$\frac{P}{P^0} = \exp\left(\frac{C}{A} - \frac{B\theta}{A} \right) \left(\frac{\xi}{\pi} \right)^{A-2} \left(\frac{\xi}{\pi} \right)^{B-1} \frac{1}{\pi} \exp\left(\frac{C}{A} - \frac{B\theta}{A} \right)$$

The parameters of π in above equations were listed in Table R15.

Table R15 Parameters for the vapor pressure equation

π_0	π_1	π_2	π_3	π_4	π_5	π_6	π_7	π_8	π_9
0.28	4.30	0.60	0.21	5.10	0.49	0.362	-4.75	-0.40	0.03

Finally, the sorption capacity of the sorbent with the salt content of x can be theoretically calculated as the following equation:

$$w = x w$$

Fig. R15 a Calculated isotherms of LiCl with the initial specific mass of 90 wt% and the experimental isotherms of Li-SHC-90 and the pure matrix. **b** Calculated theoretical isotherms of Li-SHC (90 wt% LiCl) under different temperature.

According to above theoretical modes, the calculated isotherm of pure LiCl with the initial specific mass of 90 wt% were shown in Fig. R15a. This theoretical isotherm shows high consistency with the experimental Li-SHC-90, indicating that isotherms of Li-SHC-90 sorbent and pure LiCl are essentially the same. Furthermore, the theoretical isotherms of LiCl (90 wt%) under different temperatures are shown in Fig. R15b, indicating the insensitivity to temperature and sensitivity to relative humidity. This also shows the adaptability of sorbents to different climatic conditions (even lower than 0 °C).

Note that the LiCl salt and its solution adsorption stage (salt solution zone) accounts for over 85% of the total water uptake (when RH > 60%), highly contributing to the water uptake performance. The water uptake capacity can be highly increased by working in the salt solution zone. Overall, increasing salt content and making sorbents work in the salt solution absorption zone is a common synthetic

strategy for the AWH-used sorbents. The salt content of the previous salt-embedded composite sorbents was summarized and shown in Fig. R16.

Fig. R16 Salt content of salt-based composite sorbents used in AWH field

The stability of the device and the sorbent

The stability of the material and device is indeed an important issue in a long-term application, and we comprehensively did the stability evaluations of both the device and the material.

To evaluate the stability of the sorbent, we performed a 180-hour water sorption-desorption cycling experiments using the sorbent with the area of 9 cm² inside the constant environmental chamber. The sorption and desorption condition are 65% RH, 15 °C and 15% RH, 80 °C, respectively. As shown in Fig. R17a, no obvious performance degradation was found during the sorption-desorption cycle. Note that the water uptake of the first sorption cycle is less than the remaining cycles as in the first active stage the sorbent absorbed water vapor from the total dry state.

In addition, the risk of the salt solution was tested at high RH (75% RH and 95% RH), showing that no solution leakage was observed even at RH as high as 95% (Fig. R17b). This is because the solution was totally encapsulated inside a waterproof but moisture permeable membrane with the average pore size of 10 μm. The humidity in the air can be permeable and adsorbed by the sorbent, but the generated salt solution cannot cross through the waterproof membrane to leak, which ensuring the stability and reliability of the sorbent Li-SHC. The membrane encapsulation can bring other advantages, such as avoiding the metal corrosion and preventing water quality deterioration.

Fig. R17 Stability evaluation of the sorbent. **a** 180-hour sorption-desorption cycling tests of the sorbent Li-SHC at 65% RH, 15 °C for sorption and 15% RH, 80 °C for desorption. **b** Photos of the sorbent Li-SHC after exposing to the high RH of 90% RH, showing the solution leakage was avoided by the membrane encapsulation.

To evaluate the stability of the device, we conducted a 6-day water harvesting and batch-processed water production cycling experiments under three typical climate conditions in our field test location, Lanzhou, China. The three nighttime sorption conditions are the typical climate in the spring/autumn season (15°C, 65% RH), the typical climate in the winter season (5°C, 50% RH), and the typical climate in the summer season (20°C, 75% RH). Experiments under each climate condition were conducted for 2 days, with a total experimental time of 6 days. As shown in Fig. R18a and Fig. R18b, the water production remained consistent in each climate condition, which are ca. 340 g at the spring condition, ca. 280 g at the winter condition, and ca. 425 g in the summer condition. The total water production over the 6 days is ~2115 g (Fig. R18c), showing the excellent water production performance and the adaptivity of the water harvester to different climates. As shown in Fig. R18d, the total energy consumption is 5.97 kWh, corresponding to the water production of ~355 g per kilowatt-hour.

In addition to the overall water production and energy consumption, the mass changes of the sorbents during the cycles are shown in Fig. R18e. It shows that the average water uptake of each sorbent is 2.41 g/g, 2.04 g/g, and 2.81 g/g for each climatic condition, respectively. More importantly, all sorbents have no obvious mass loss after these cycles, showing excellent stability throughout the cyclic tests with variable working conditions. Detailed information on the temperature variation of each component of the water harvester can be found in Fig. R18f.

Fig. R18 Stability evaluation of the device at various simulated climate conditions. **a** Photos of the produced freshwater on each day. **b** Collected freshwater amounts of each cycle and each day. **c** Overall water production during the cyclic tests. The insert shows the photos of the overall collected water on each day **d** Overall energy consumption during the cyclic tests. **e** Mass changes of each piece of the sorbent during the 6-day water capture-release cycling tests. **f** Temperature variation of the heating plate, sorbent, chamber, and two measuring points on the condenser. The measuring points of the sorbent moved due to released vapor turbulence, resulting in the measured temperature fluctuations.

Changes made:

We added the description of the **stability** evaluation on **Page 8 & 14** and related **SI**, highlighted in red.

“Sorption-desorption cycles performed over 186 hours confirmed the stable equilibrium and dynamic sorption performance without degradation and LiCl leakage (Fig. 2d and Supplementary Fig. 15).”

“Finally, the stability of the device was evaluated by conducting the 6-day water harvesting and batch-processed water production cycling experiments under various simulated climate conditions, further demonstrating the stability of the bulk sorbents and the strong environmental adaptability of the water harvester (Supplementary Fig. 28).”

We updated the experimental methods related to the stability evaluation in **Methods** part, highlighted in red.

“The sorption-desorption cyclic experiments were conducted to evaluate the stability of LiSHC. The dehydrated sample with an area of 3 x 3 cm² firstly absorbed water vapor at 15°C and 65% RH inside the constant climate chamber (KMF-115, Binder) for 12 hours. After that, the environment was changed to the desorption condition (80°C, 15% RH) for 6 hours. The sorption-desorption cycles were repeated ten times, which took over 10,000 mins. The weight change of the sample was recorded by the analytical balance. The stability and adaptability of the water harvester were evaluated by the 6-day water collection experiments under three various simulated climatic conditions. Detailed information can be found in Supplementary Information.”

We added the explanation about the water uptake contribution on **Page 5 & 6** and related **SI**, highlighted in red.

“The abundant micropores of carbon fibers provide ultra-high specific surface areas (Supplementary Fig. 1), and their physical entanglement and numerous channels enhance the hydrophilicity (Supplementary Fig. 2) and capillary force of the felt (Supplementary Fig. 3). The water uptake isotherm of the pure porous matrix (Supplementary Fig. 4) demonstrates the micropore filling at the low RH (< 30%) and an increased water uptake at higher RH owing to the multilayer adsorption and capillary condensation in mesoporous, but the relatively low

water adsorption capacity ($< 0.1 \text{ g g}^{-1}$ below 30% RH) can hardly be suitable for water harvesting in arid climates. Therefore, hygroscopic nanoscale LiCl salts were uniformly loaded to the porous matrix by vacuum impregnation and mild heating methods to promote the water harvesting capacity in a wide working range of RHs. Meanwhile, the active carbon fiber felt serves as the matrix to support and disperse LiCl to alleviate the salt agglomeration and salt solution leakage. After LiCl loading, the totally dried sorbent becomes less flexible and shows a higher mechanical strength than the pure matrix. However, the flexibility of the sorbent can be recovered after sorption (Supplementary Fig. 5).

...

To reveal the sorption mechanism of Li-SHC, the phase diagram and the theoretical isotherm of pure LiCl were calculated (Supplementary Information Section 3) and presented in Supplementary Fig. 12. Obviously, the isotherm of Li-SHC shows highly similar isotherm of pure LiCl, which indicates that the equilibrium water uptake of Li-SHC is mainly determined by the sorption capacity of LiCl salt. Or to say, the porous matrix has a negligible contribution to the equilibrium water uptake. The water uptake of Li-SHC shows the multi-step mechanism as demonstrated in Fig. 2a: 1. Chemisorption of anhydrous LiCl below the deliquescence RH (11% RH at 30 °C); 2. Deliquescence of hydrous LiCl ($\text{LiCl}\cdot\text{H}_2\text{O}$) and the saturated solution forms 3. Absorption of water vapor in saturated/concentrated LiCl solution. Note that over 85% of total water uptake was attributed by the adsorption of water vapor in LiCl solution when sorption at $\text{RH} > 60\%$, thus the third step highly contributes to the overall water uptake capacity.”

We added a **dynamic sorption** model to evaluate the dynamic characteristics of the sorbent. The information was updated on **Page 6**.

“The linear drive force (LDF) model was used to quantitatively evaluate their dynamic sorption characteristics (Supplementary Information Section 2). The sorption rate coefficients k_{LDF} indicates a strong positive correlation between the salt content and the sorption rate (Supplementary Fig. 10). This variation of dynamic sorption characteristic owes to the mass transfer enhancement effect, provided by the matrix, as it leverages the effect of additional pores and channels for increased sorption interfacial area. However, with the increased salt content, LiCl crystals form not only on the fiber pores but also on the surface and between the gaps of fibers (SEM images see Supplementary Fig. 11), which inevitably reduced the vapor transport channels and the sorption dynamics. Note that although reducing the salt loading can increase the sorption

dynamics, the absolute amount of water uptake decreases simultaneously. The desired salt content in practical water harvesting applications can be optimized by matching the duration of a sorption cycle with the required equilibrium duration, indicating no waste of both the sorbent material and the operating time.”

Another concern is about Fig. S23 and the closeness of the results achieved in this manuscript and those in Xu et al. *Energy & Environmental Science* (2021).

As already discussed, the similarity is due to the same LiCl for both materials. Fig. S24 shows an advantage of the device in the present manuscript that is due to the absence of the PV panel and of the fan, being the device of this manuscript less sophisticated (a no-fan and grid-connected, optional PV connection but not experimented).

However, the main factor contributing to the unrivalled amount of water produced is the periodic replacement of the sorption material. This is a practice common to both the present manuscript and of Xu et al. *Energy & Environmental Science* (2021), which is from the same senior author. All other compared systems do not use to replace manually the sorption material, hence achieving lower water yields.

Response:

We agree with your comments about our device. As we mentioned in the former part, we used controlled natural air flow that is a consequence of enhanced temperature and vapor concentration gradients instead of relying on the forced air flow that is achieved by the fans. Our main design goal was to ensure the portability and stability by designing simple water harvester without excessive amount of auxiliary devices, such as fans and else. Moreover, the field test was demonstrated by a grid-connected system, even though we have calculated the required area of PV modules and the required battery capacity.

As for the operation strategy, the periodic replacement of sorbents (batch-process) leads to enhanced water yield, achieving over three-hundred-gram daily water production. The reason why we applied the batch operation mode is due to the large mismatch between the sorption and desorption rates. The maximum desorption rate is ~10 times higher than the sorption rate. The batch-process mode allowed the desorption rate to be quasi-continuously maintained at highest rate by alternating sorbent pieces, which yielded larger amount of water in certain period. However, if the water harvesting operation is limited to only one sorption-desorption cycle a day (no sorbent replacement), the water

production is 29.2 to 63.7 mL. The produced water range is mainly affected owing to different desorption times and numbers of layers, as shown in Supplementary Information Section 4. In this case, the calculated water production per gram of sorbent was 0.77 to 1.68 $\text{g}_{\text{water}}/\text{g}_{\text{sorbent}}/\text{day}$, illustrating that our device with corresponding sorbents remains competitive in comparison to other water harvesters that usually have a lower than 1.0 $\text{g}_{\text{water}}/\text{g}_{\text{sorbent}}/\text{day}$ water yield, as shown in Supplementary Figure 29c.

Changes made:

We highlighted and re-organized this part. The related information was added in *SI: Supplementary Section 4. Optimization of Water Harvester & Supplementary Section 8. Performance Evaluation.*

Finally, there are some minor points that require the authors' attention:

- 1) The English shows sometimes missing "s", "these" instead of "this", "Contrary" instead of "Conversely", etc..

Response:

We are sorry for our incorrect writing. All related texts have been checked and corrected.

- 2) "Kinetics" is improperly used in the manuscript to refer to the dynamic behaviour of a sorbet structure. Kinetics is the diffusion in the material, which is a property of the material distinct from other properties such as the thermal conductivity for example. The authors should use the term dynamics in experiments in which the heat and mass transfer affect each other (typically when the amount of material is significant, usually in the order of mg or more). **Response:**

We are sorry for confusing the physical meanings of "kinetics" and "dynamics". We have made correction in the revised manuscript.

- 3) Row 268 reports "Section S3" while it should be "Section S4"

Response:

We carefully checked the re-submitted version and tried our best to avoid this kind of mistake.

Other changes made:

Upon checking the manuscript again, we have found several technical errors that have been resolved in the latest version of the manuscript. Some parts have been deleted due to the word count limitation of the main text. We also updated the latest information in this AWH field to enhance the readability

of our manuscript. Additionally, the language and structure of the manuscript is further polished. All changes were shown in following:

“Among all atmospheric water harvesting (AWH) technologies, sorption-based AWH (SAWH) shows high adaptivity and fewer limitations on environmental conditions, as sorbents have a strong affinity to capture water from air at a wide range of RH (~10-100% RH) and release it due to water vapor partial pressure difference.”

“Therefore, the cheap and industrially mass-produced raw materials and the simple production processes make the salt-based composite sorbents highly suitable for mass production and show great market potential.”

“Meanwhile, combining AWH systems with other applications, such as thermal management, energy generations or storage, and agriculture, can also be regarded as the methods to improve the overall production/efficiency of the AWH-related systems.”

“Contrary, active sorption-based water harvesters that rely on electric or thermal energydriven sorption processes as input are getting more attention owing to their stable water productivity and significantly higher yield, whereas the weight and size of these harvesters are usually unacceptable especially for portable SAWH applications.”

“Coupled with the excellent performance of Li-SHC and the advanced thermal strategies of the device, the portable water harvester achieved the water productivity of up to 312 mL day⁻¹, 55.66 g_{water} L_{device}⁻¹ day⁻¹, and 97.40 g_{water} kg_{device}⁻¹ day⁻¹, by applying eight desorption cycles per day, in the real semi-arid environment (Lanzhou, China) with the extreme low RH of ~15%. Moreover, the harvester with lightweight sorbents can be easily deployed by a single person.”

“The cut-section scanning electron microscope (SEM) image (Fig. 1b) and the energydispersive X-ray spectroscopy (EDX) mappings of Li-SHC (Fig. 1c) confirmed the LiCl crystals were successfully loaded on the surface of fibers and between the channels formed by the physical entanglement. When exposed to the moist air, the water molecules can be absorbed by LiCl crystals on the surface of the fibers (Fig. 1d), then diffuse into the porous matrix, forming concentrated LiCl solution. The volume of LiCl solution is expanded during the following

adsorption process, and the formed solution is stored inside the hydrophilic and highly porous matrix with strong capillary force.”

“Therefore, two mismatches need to be particularly considered to pursue maximum water production yield if applying the sorbent to real devices: 1. the sorption performance of real bulk sorbents versus the milligram-level test samples; 2. the slow water sorption rate and the relative fast desorption rate. Fortunately, although a slower sorption dynamic limits the sorption performance of Li-SHC within 12 hours, Li-SHC still shows a large amount of water release (2.37 g g^{-1}) in one water harvesting cycle under this actual situation, which benefits from the strong water affinity of LiCl salts, the high salt content of sorbents, and the better thermal conductivity of the carbon-based matrix. Also, to ensure the quasi-continuous water collection during the daytime, a batch-process operation mode is proposed to maintain a fast desorption rate, which is expected to improve diurnal water collection rates.”

Finally, we authors thank the editor and reviewers again for your valuable comments and constructive suggestions!

References

1. Meyers LD, Hellwig JP, Otten JJ. *Dietary reference intakes: the essential guide to nutrient requirements*. National Academies Press (2006).
2. Gleick PH. Basic Water Requirements for Human Activities: Meeting Basic Needs. *Water International* **21**, 83-92 (1996).
3. Humphrey JH, *et al.* The potential for atmospheric water harvesting to accelerate household access to safe water. *The Lancet Planetary Health* **4**, e91-e92 (2020).
4. Stoeckli HF, Kraehenbuehl F, Ballerini L, De Bernardini S. Recent developments in the Dubinin equation. *Carbon* **27**, 125-128 (1989).
5. <https://sage.nelson.wisc.edu/data-and-models/atlas-of-the-biosphere/mapping-thebiosphere/ecosystems/average-annual-relative-humidity/>. *Climate Research Unit, Univ of East Anglia*, (1999).
6. Wahlgren RV. Atmospheric water vapour processor designs for potable water production: a review. *Water Research* **35**, 1-22 (2001).
7. Tu Y, Wang R, Zhang Y, Wang J. Progress and Expectation of Atmospheric Water Harvesting. *Joule* **2**, 1452-1475 (2018).
8. Hanikel N, *et al.* Rapid Cycling and Exceptional Yield in a Metal-Organic Framework Water Harvester. *ACS Cent Sci* **5**, 1699-1706 (2019).
9. Wang W, Xie S, Pan Q, Dai Y, Wang R, Ge T. Air-cooled adsorption-based device for harvesting water from island air. *Renewable and Sustainable Energy Reviews* **141**, (2021).
10. Xu J, *et al.* Ultrahigh solar-driven atmospheric water production enabled by scalable rapid-cycling water harvester with vertically aligned nanocomposite sorbent. *Energy & Environmental Science* **14**, 5979-5994 (2021).
11. Aristov YI, Restuccia G, Cacciola G, Parmon VN. A family of new working materials for solid sorption air conditioning systems. *Applied Thermal Engineering* **22**, 191-204 (2002).
12. Aristov YI. Composite Sorbents of Water Vapor: Effect of a Confined Salt. In: *Nanocomposite Sorbents for Multiple Applications*. Jenny Stanford Publishing (2020).
13. Aristov YI. Composite Sorbents of Water Vapor: Effect of a Host Matrix. In: *Nanocomposite Sorbents for Multiple Applications*. Jenny Stanford Publishing (2020).
14. Yan S, *et al.* Effect of fiber content on the microstructure and mechanical properties of carbon fiber felt reinforced geopolymer composites. *Ceramics International* **42**, 7837-7843 (2016).
15. He P, *et al.* Effects of fiber contents on the mechanical and microwave absorbent properties of carbon fiber felt reinforced geopolymer composites. *Ceramics International* **44**, 10726-10734 (2018).
16. Lin T, Jia D, He P, Wang M. In situ crack growth observation and fracture behavior of short carbon fiber reinforced geopolymer matrix composites. *Materials Science and Engineering: A* **527**, 2404-2407 (2010).
17. Lawson KW, Lloyd DR. Membrane distillation. *Journal of Membrane Science* **124**, 1-25 (1997).
18. Wang W, *et al.* Integrated solar-driven PV cooling and seawater desalination with zero liquid discharge. *Joule* **5**, 1873-1887 (2021).

19. Baghel R, Upadhyaya S, Singh K, Chaurasia SP, Gupta AB, Dohare RK. A review on membrane applications and transport mechanisms in vacuum membrane distillation. *Reviews in Chemical Engineering* **34**, 73-106 (2018).
20. Alkudhiri A, Darwish N, Hilal N. Membrane distillation: A comprehensive review. *Desalination* **287**, 2-18 (2012).
21. Sircar S. Linear-driving-force model for non-isothermal gas adsorption kinetics. *Journal of the Chemical Society, Faraday Transactions 1: Physical Chemistry in Condensed Phases* **79**, 785-796 (1983).
22. Sircar S, Hufton JR. Why Does the Linear Driving Force Model for Adsorption Kinetics Work? *Adsorption* **6**, 137-147 (2000).

Reviewers' Comments:

Reviewer #1:

Remarks to the Author:

With their careful revisions, the authors have well addressed my comments. This paper can be accepted now for publication in Nature Communications.

Reviewer #2:

Remarks to the Author:

The authors have analyzed the manuscript in great details regarding to comments of reviewers, and all comments have been addressed in the revised manuscript. Therefore, I think the revised manuscript is suitable for publication in Nature Communications.

Reviewer #3:

Remarks to the Author:

This reviewer cannot see in the new version of the manuscript any substantial change that would encourage publication in its current version. The answers received fall short in a number of important fundamental aspects that should be better checked by the authors. I encourage the authors to consider genuinely the comments and answer in a more direct and less questionable way.

My first concern is still on the novelty of this work. In their reply, the authors identify two novelties that are (1) Dynamics of the material in a real device; (2) Optimisation of the device.

Previously published research involved device-level optimizations and/or field tests in locations different from the one of the present manuscript.

Authors claim this manuscript is distinct from previously published work since (i) previously published work has cycled the material on multiple consecutive days while water is mainly produced in the first cycle; (ii) previously published work used a different material in a different location producing significantly less water per day.

As far as the first point is concerned, authors haven't cycled their material, replacing daily the used material capsule with a fresh one. This is one shortcoming of the work compared to previous publications. It is well known that salt-supported materials give their best at their first cycle, significantly underperforming in all subsequent cycle, which is still the real open challenge. Unfortunately, the challenge remains unaddressed.

As far as the second point is concerned, the manuscript looks in some aspects anticipated. The current manuscript shows field tests in a location where the day/night change of RH is larger than in previous location (Shanghai). However, the new material only marginally improve the water production (up to a factor of 1.18 improvement), providing the reviewer with the impression that a number of elements of this manuscript have been already seen in other previously published work. This undermines the novelty content and would suggest a different editorial collocation for this research.

My second concern is on the claim. The changes made to highlight that the new device is simpler than the previous still give the impression that "ultra-high 97.40 gram per kilogram of device (gwater kgdevice-1 day-1)" is obtained thanks to the material, while in reality the record value is obtained thanks to a reduction of auxiliary device components. This should be explicitly and more honestly stated in the manuscript.

Third concern is on dynamics.

My first note is that the fittings in Fig.R15 are incorrect. The LDF model supposes isothermal material, while the heat of adsorption present in the data would require to suit the LDF model with a valid energy balance equation.

However, I understand and share the point of the authors on the improvement of the dynamics but still the salt would deliquesce in the test conditions, collecting in the bottom of the capsule. Encapsulation in the membrane do not prevent the salt leaks out of the material and only ensures its confinement in the capsule. It is likely the salt would move to the bottom of the membrane capsule leaving the support. Opening the capsule and showing that salt is not collected in the bottom would reassure all concerns on the material stability. Furthermore, day/night adsorption systems have such a long dynamics that equilibrium overshadows practically diffusion.

Last concern is on material stability. The authors prepared FigR17 and FigR18 to answer. In FigR17 there is a mismatch between the RH in picture (65%) and the one stated in the caption (90%). More importantly, as explained above, the membrane capsule should be opened to check whether the salt remained in the initial position or settled in the bottom of the capsule. This would definitely clarify on stability.

FigR18 is provided but does not add much to the discussion. This figure shows system stability (as pointed by the authors although not asked by the reviewer) and not material stability (as asked by the reviewer) since material is replaced at each cycle.

Reviewer #1

With their careful revisions, the authors have well addressed my comments. This paper can be accepted now for publication in Nature Communications.

Responses:

We really appreciate the Reviewer for the careful review and helpful comments. We thank your positive assessment of the revised manuscript.

Reviewer #2

The authors have analyzed the manuscript in great details regarding to comments of reviewers, and all comments have been addressed in the revised manuscript. Therefore, I think the revised manuscript is suitable for publication in Nature Communications.

Responses:

We are glad that the Reviewer believes all the previous comments have been well addressed. We appreciate the Reviewer for your valuable comments to help us improve the quality of our work.

Reviewer #3

This reviewer cannot see in the new version of the manuscript any substantial change that would encourage publication in its current version. The answers received fall short in a number of important fundamental aspects that should be better checked by the authors. I encourage the authors to consider genuinely the comments and answer in a more direct and less questionable way.

Responses:

We thank the reviewer for the comments and regret that the additional experiments and possibly complicated explanations in the last responses did not make the reviewer satisfied but weakened the focus and clarification of our work. We would like to briefly summary the main points that we have made the efforts to respond to the reviewer's comments in the last responses.

1. We highlight the three advantages and novelties of our work by analysing the two articles mentioned by reviewers. Based on that, the advantages and main novelties include 1) the stable, low-cost, scalable, and high-performance composite sorbents, 2) the advanced thermal management of the water harvester, and 3) the proposal of a novel operation strategy using 8-cycle batch-processed water release mode.
2. We provided the water vapor sorption model regarding the material to illustrate the importance of material-level optimization in terms of equilibrium and dynamics.
3. We performed the cyclic sorption-desorption experiments with the material and continuous 6-day water production experiments using the water harvester under various typical climates to demonstrate both materials and system stability.

We hope that this comprehensive but concise summary would help reviewers to get the main points of our work and the responses. In the following sections, we would address the reviewer's concerns point by point.

My first concern is still on the novelty of this work. In their reply, the authors identify two novelties that are (1) Dynamics of the material in a real device; (2) Optimisation of the device.

Previously published research involved device-level optimizations and/or field tests in locations different from the one of the present manuscript.

Authors claim this manuscript is distinct from previously published work since (i) previously published work has cycled the material on multiple consecutive days while water is mainly produced in the first cycle; (ii) previously published work used a different material in a different location producing significantly less water per day.

As far as the first point is concerned, authors haven't cycled their material, replacing daily the used material capsule with a fresh one. This is one shortcoming of the work compared to previous publications. It is well known that salt-supported materials give their best at their first cycle, significantly underperforming in all subsequent cycle, which is still the real open challenge. Unfortunately, the challenge remains unaddressed.

Responses:

We thank the reviewer for the comments. In the last revision, we provided an extensive set of information about the novelty of our work with two points that have been listed above, which are

- 1. the device-level optimisation using advanced thermal solutions to reduce the complexity of the device and maintain the portable advantage.**
- 2. materials optimisation, provided by the sorption isotherm and dynamic model based on the working sorption duration and relative humidity.**

Beyond these, we believe an important novelty point related to **the operation strategy** has not been noticed by the reviewer.

Figure R1 Illustration of different operation strategies used for atmospheric water harvesting.

a Traditional single water capture-release cycle in a diurnal cycle. **Blue line – RH variation.** Black line – Water sorption amount. **Yellow line – Water production over the whole day.** As the desorption rate is much higher than the adsorption rate, much of the daytime (around 10 hours) is wasted.

b Derivative water uptake of our materials showing the mismatches between sorption/desorption rates.

c Semi-continuous water capture-release mode. **Blue line – RH variation.** Black line – Water sorption amount. **Yellow line – Water production over the whole day.** The temperature variation caused by the switch of sorption and desorption processes causes waste of energy, meanwhile, the low RH during the daytime causes inefficient water uptake.

d Novel batch-processed alternating mode used in our paper. **Blue line – RH variation.** Black line – Water sorption amount. **Yellow line – Water production over the whole day.** With this, we can make full use of the high humidity at night to fully absorb water by multiple pieces of the material and desorb them during the day using an alternate mode. The sorbents can be reused in the following day without replacing them.

The operation mode of the most recent research in the AWH field is simple – single water capture-release cycle in a diurnal cycle (Figure R1a). However, this cycle mode is inefficient, largely because of the apparent mismatch between the sorption/desorption dynamics – **the desorption process is much faster than the sorption process** (up to 14 times at 15%RH that

reported by EES ¹ and 10 times for our case) (Figure R1b or Fig. 2f in main text). This single sorption-desorption mode wastes most of the daytime (9-11 hours). A few studies noticed the mismatch and tried to solve it by applying the semi-continuous AWH ² (for example, ACS Central Science ³ and EES ¹ papers mentioned by the reviewer) – The sorbents sorb and desorb water **in a cycling manner** throughout the whole day including **the daytime** (Figure R1c). However, RH at daytime is always much lower than at night-time because of the diurnal temperature difference, especially in arid and semi-arid regions. The low RH at daytime caused the low water uptake of the sorbent, thus the water production during the daytime is also limited, although the additional time and energy consumption wastage caused by the heating and cooling processes due to the switch between the adsorption and desorption modes is not considered.

Therefore, there should be an operation strategy that can **not only make use of the high RH during the night, but also maintains the desorption rate as high as possible during the entire daytime**. Based on this, we proposed a novel operation mode - **multiple sorbents sorb water vapor at night with high RH with the subsequent alternated desorption process during the day**. (Figure R1d). This method avoids conducting the inefficient sorption process under low humidity during the day, thus guaranteeing high water uptake of each sorbent as well as high water release yield in a single diurnal cycle. This is an important reason why our work realizes an exceptional water production yield, and together with the advanced thermal design, making our work distinct from the previous work. To show our proposed operation mode more clearly, some information found in Figure R1 was added to the Supplementary Information Fig. 28.

Particularly, we would like to stress out that the adoption of this strategy does not mean that our materials are less stable in cycling. In fact, we **have not daily replaced** the used materials with fresh one, as the reviewer said. Instead, as shown in the **device stability** experiment, we used the **same** eight pieces of sorbents throughout the 6-day sorption-desorption process (Figure R2 or Fig. R18e in the last responses letter). It is noted that nearly all sorbents obtained the same water uptake for the same working conditions within two continuous days, and after changing the working conditions it still maintains good stability and works well in the changed conditions. After six days of experiment, the 8 pieces of materials are completely dried and

have not shown any mass (salt) loss. Based on this, we showed strong stability and environmental adaptability of both the materials and the device. Additionally, we will have more detailed discussion on the stability of materials in the last part of the response letter.

Figure R2 Mass change variations of the eight pieces of the sorbents during the 6-day cycles. The same water uptake was obtained for the same sorption conditions and no mass (salt) loss was observed after 6-day cycles. Day 1-2: Sorption at 65% RH, 15 °C. Day 3-4: Sorption at 50% RH, 5 °C (Winter); Day 5-6: Sorption at 75% RH, 20 °C.

As far as the second point is concerned, the manuscript looks in some aspects anticipated. The current manuscript shows field tests in a location where the day/night change of RH is larger than in previous location (Shanghai). However, the new material only marginally improve the water production (up to a factor of 1.18 improvement), providing the reviewer with the impression that a number of elements of this manuscript have been already seen in other previously published work. This undermines the novelty content and would suggest a different editorial collocation for this research.

Responses:

Thank you for your comments. Honestly, as the reviewer said, the sorbent used in our work indeed improve only 18% equilibrium water uptake (sorption isotherm) compared to the EES paper, but water uptake of the milligram-scale material sample that is tested in the laboratory

cannot reflect the water production in real device and real scenario, such as proposed in the manuscript. Actually, the daily water production is far away from the sorption capacity shown in the sorption isotherm, which is still a real challenge. This point of view is emphasized many times in our manuscript, and this is why we made efforts to the heat and mass transfer enhancements of the device, as well as the operation strategy. As mentioned above, by analysing the mismatch between sorption and desorption rates, the advanced operation strategy is newly proposed and its effect is remarkable, where multiple sorbents sorb water vapor during the night and alternatively desorb water during the day. With the help of material-level and system-level optimisations, the portable water harvester with the small volume and light weight achieved the water production over 300 mL per day.

Certainly, the two papers that the reviewer mentioned in the last comments are definitely very important papers. And praiseworthy, they are the very few works that involve the device-level optimisations and/or field tests. We have learned much from their work, but our work is still distinct in terms of **operation modes** (as analysed above) and **how we design the material and systems from a holistic perspective**. Together with these two studies, we hope that the design approach from material, device, to operation strategy, could stimulate more interest for future research in AWH fields. We further emphasized the contribution of these two works in the *introduction* section, as following “For example, multi water harvesting cycles over the whole day were implemented in a MOFs water harvester, achieving the water productivity of $0.7 \text{ L kg}_{\text{MOF}}^{-1} \text{ day}^{-1}$ in the real desert climate. Besides, another water harvester with eight water capture-release cycles recently achieved ultrahigh outdoor water productivity of $1.05 \text{ L}_{\text{water}} \text{ kg}_{\text{sorbent}}^{-1} \text{ day}^{-1}$ only driven by natural sunlight.”

My second concern is on the claim. The changes made to highlight that the new device is simpler than the previous still give the impression that “ultra-high 97.40 gram per kilogram of device ($\text{g}_{\text{water}} \text{ kg}_{\text{device}}^{-1} \text{ day}^{-1}$)” is obtained thanks to the material, while in reality the record value is obtained thanks to a reduction of auxiliary device components. This should be explicitly and more honestly stated in the manuscript.

Responses:

Thanks for your suggestions and comments. We agree with the reviewer's statement that the high water production yield in the portable water harvester is thanks to the reduction of auxiliary device components by enhancing the heat and mass transfer inside the water harvester and employing the novel operation strategy. To discuss this part more clearly, we checked all the manuscript related to the statement. We have deleted or modified the expression on the evaluation of the water harvesting performance regarding the weight and volume of the water harvester. The modifications were included in the following:

1. In the *Introduction* part, we added “the batch-processed operation strategy”, deleted “ $55.66 \text{ g}_{\text{water}} \text{ L}_{\text{device}}^{-1} \text{ day}^{-1}$, and $97.40 \text{ g}_{\text{water}} \text{ kg}_{\text{device}}^{-1} \text{ day}^{-1}$ ”, and added “with the volume of 5.6 L and the weight of 3.2 kg” to highlight the importance and the effect of the operation strategy on the portability of the water harvester and overall water production yield. The sentence was changed to “Owing to the excellent performance of Li-SHC, the advanced thermal strategies of the device and the batch-processed operation strategy, the portable water harvester with the volume of 5.6 L and the weight of 3.2 kg achieved water productivity of up to $311.69 \text{ g day}^{-1}$ and $1.09 \text{ g g}_{\text{sorbent}}^{-1} \text{ day}^{-1}$, by applying eight desorption cycles in the real semi-arid environment (Lanzhou, China) with the extreme low RH of $\sim 15\%$.”
2. In the *Multicycle AWH cycles field test in the semi-arid region* part, when discussing the metrics of the **sorbent**, we deleted the statement “showing 1.09 grams of obtained water per gram of sorbent per day ($\text{g}_{\text{water}} \text{ g}_{\text{sorbent}}^{-1} \text{ day}^{-1}$) owing to the high sorption capacity of sorbent” in order to not mislead the reader. Meanwhile, considering the multifarious sorbents, structures, and energy sources involved in water harvesters, we deleted the discussion of the metrics comparison. Instead, we more focused on the performance and portability of our water harvester. We deleted the Figure about the evaluation of the presented water harvester in terms of comprehensively evaluated metrics based on existing sorbents and corresponding devices. The related Supplementary Figure, Table, and discussions were modified.
3. In the *Abstract* and *Conclusion* part, we added “with the volume of 5.6 L and the weight of 3.2 kg” as well as “Together with the proposed novel eight-cycle batch alternating desorption mode to maintain the high water desorption rate during the daytime, ...” to highlight that the high water production was achieved due to the reduction of auxiliary device components and the employment of the novel operation strategy.

Third concern is on dynamics.

My first note is that the fittings in Fig.R15 are incorrect. The LDF model supposes isothermal material, while the heat of adsorption present in the data would require to suit the LDF model with a valid energy balance equation. However, I understand and share the point of the authors on the improvement of the dynamics but still the salt would deliquesce in the test conditions, collecting in the bottom of the capsule. Encapsulation in the membrane do not prevent the salt leaks out of the material and only ensures its confinement in the capsule. It is likely the salt would move to the bottom of the membrane capsule leaving the support. Opening the capsule and showing that salt is not collected in the bottom would reassure all concerns on the material stability.

Responses:

We thank for your comments. The LDF model is indeed derived from the isothermal sorption condition, but this model has fitted well accordingly with $R^2 > 0.95$ in all accounts when modelled over the entire data range due to the uniform and nearly constant temperature of the sorbent in the environmental chamber. Such temperature resulted due to the used thin carbon-supported materials and efficient heat transfer between sorbent and external environment. We noticed that the LDF model has been widely used in studying the dynamic water uptake of the sorbent in AWH fields.^{4, 5, 6, 7}

Last concern is on material stability. The authors prepared FigR17 and FigR18 to answer.

In FigR17 there is a mismatch between the RH in picture (65%) and the one stated in the caption (90%). More importantly, as explained above, the membrane capsule should be opened to check whether the salt remained in the initial position or settled in the bottom of the capsule. This would definitely clarify on stability.

FigR18 is provided but does not add much to the discussion. This figure shows system stability (as pointed by the authors although not asked by the reviewer) and not material stability (as asked by the reviewer) since material is replaced at each cycle.

Responses:

We thank the reviewer for the comments. In the Fig. R17a, the sorption RH is 65%, which is consistent with the caption. In the Fig. R17b, we are sorry for the typing errors. Now we have corrected the text in the caption, thank you so much for pointing this out! It is consistent with the real tested high RHs (95% and 75% RH). As following, we provide our reply point by point according to the reviewer's views.

“It is well known that salt-supported materials give their best at their first cycle, significantly underperforming in all subsequent cycle, which is still the real open challenge. Unfortunately, the challenge remains unaddressed.”

As for the stability concern, we understand that, for a long time in the past, the salt-based composite sorbents did suffer from long-term stability when exposed to the high relative humidity for a long time, which is a side effect of the deliquescence of hygroscopic salt to increase sorption capacity⁸. The direct reason why such situation, described by the reviewers, is happening

“best at their first cycle, significantly underperforming in all subsequent cycle”

is that **the generated salt solution after exposed to high relative humidity for a long duration was leaked from the matrix**⁹. Once discharged from the sorbent material, the salt solution cannot be taken back to the inner part of the materials to execute its strong hygroscopicity in the next cycle anymore. Therefore, to ensure the stability of the equilibrium/dynamic water uptake, the appropriate strategy is to avoid the salt solution leakage from the entire sorbent material.

Recently, the solution leakage problem has been successfully avoided by a novel method using an encapsulation procedure, where the hygroscopic components are embedded into a hydrophobic but moisture permeable layer. This layer ensures that the salt solution (hygroscopic components) can be retained inside the entire material, with low influence on the vapor permeability and sorption. The layer can be composed of the fibres produced by electrospinning⁹, hydrophobic-modified cotton fabric¹⁰, and even the candle soot layer¹¹. The

novel materials with the hydrophobic but moisture permeable layer are treated as a whole, describing as the composite sorbents. We also encourage the reviewer to treat our developed sorbent as a whole composite.

In this study, we selected the commercial PTFE membrane with hydrophobic and moisture permeable functions to encapsulate the hygroscopic components to form the whole composite material¹². The commercial membranes were chosen to ensure the scalability and low costs of the materials. As a result, the risk of the solution leakage when the sorbent exposed to high RH environment for too long time is thus avoided. In the Fig. R17a (Figure R3), the 180-hour water sorption-desorption cycling experiments using the sorbent with the area of 9 cm² inside the constant environmental chamber have been performed. No obvious changes of **both water uptake amount and dynamic** among each cycle are observed. Also, in Fig. R17b, the photos showed that no solution leakage occurred when exposed to the high RHs (75% RH and 95% RH).

Figure R3 Cycling performance of Li-SHC. The sorption and desorption are performed at 65% RH, 15 °C and 15% RH, 80 °C, respectively.

“FigR18 is provided but does not add much to the discussion. This figure shows system stability (as pointed by the authors although not asked by the reviewer) and not material stability (as asked by the reviewer) since material is replaced at each cycle.”

Although the reviewer hasn't asked for the stability of the device, in the Fig. R18, the experiments about the device stability also demonstrate the stability of the materials, because eight pieces of sorbents were repeatedly used throughout the six-day cycle under three various climate conditions instead of replacing them every day. The same water uptake amount was recorded when it was exposed to the same climate conditions, as well as no mass (salt) loss occurred after the whole cycles. We believe the two experiments showed the strong stability and environmental adaptability on both the materials and the device levels.

“As explained above, the membrane capsule should be opened to check whether the salt remained in the initial position or settled in the bottom of the capsule. This would definitely clarify on stability.”

Thank you very much for this suggestion to evaluate the stability of the material. This is really a good point which need careful investigations and can directly show whether the salt solution is leaked or not. Although we encourage the reviewer to treat the developed encapsulated materials as a whole composite sorbent, we would like to follow your suggestions to open the encapsulation and strip the membrane to show the state of the membrane after 180-hour sorption-desorption cycles.

To eliminate the effects of gravity, we stripped the membrane and split it into two pieces. As shown in Figure R4, one is the top-inner side of the membrane, and the other is the bottom-inner side of the membrane.

Figure R4 Illustration of the positions of the stripped membrane.

Figure R5 Images of the stripped membrane captured by the optical microscope.

We pasted the stripped membranes onto the glass slides and placed them under an optical microscope. As shown in Figure R5, no obvious salt particles were found on the surface of both the top-inner and bottom-inner sides of the stripped membrane. To show the state of the fibres more clearly and find out about the existence of salt particles, the magnified images were further captured. Interestingly, no salt particles or droplets were found on the PTFE membrane but on the fibre fragments. It seems that the salt particles were carried down by the peeled fibres instead of the membranes. This result could be due to the significant differences of the hydrophilicity and hydrophobicity of the components. The PTFE membrane has superhydrophobic characteristic ($CA=170^\circ$, Supplementary Figure 8), and the fibre matrix is superhydrophilic ($CA=0^\circ$, Supplementary Figure 2) with strong capillary force (Supplementary Figure 3), resulting in the significant difference of the energetic barrier of heterogeneous nucleation. Therefore, the salt solution tends to be formed inside the hydrophilic matrix during the desorption and the salt crystallization.¹³ More importantly, owing to the aforementioned interactions between the solution, matrix and the membrane, the solution retains inside the matrix during the sorption-desorption cycling.

The related figures and discussion were added to Supplementary Figure 15.

References

1. Xu J, *et al.* Ultrahigh solar-driven atmospheric water production enabled by scalable rapid-cycling water harvester with vertically aligned nanocomposite sorbent. *Energy & Environmental Science* **14**, 5979-5994 (2021).
2. Poredoš P, Shan H, Wang C, Deng F, Wang R. Sustainable water generation: grand challenges in continuous atmospheric water harvesting. *Energy & Environmental Science*, (2022).
3. Hanikel N, *et al.* Rapid Cycling and Exceptional Yield in a Metal-Organic Framework Water Harvester. *ACS Central Science* **5**, 1699-1706 (2019).
4. Logan MW, Langevin S, Xia Z. Reversible Atmospheric Water Harvesting Using Metal-Organic Frameworks. *Scientific Reports* **10**, 1492 (2020).
5. Legrand U, Girard-Lauriault P-L, Meunier J-L, Boudreault R, Tavares JR. Experimental and Theoretical Assessment of Water Sorbent Kinetics. *Langmuir* **38**, 2651-2659 (2022).
6. Zhou X, Lu H, Zhao F, Yu G. Atmospheric Water Harvesting: A Review of Material and Structural Designs. *ACS Materials Letters* **2**, 671-684 (2020).
7. Sircar S, Hufton JR. Why Does the Linear Driving Force Model for Adsorption Kinetics Work? *Adsorption* **6**, 137-147 (2000).
8. Gordeeva LG, Aristov YI. Composites 'salt inside porous matrix' for adsorption heat transformation: a current state-of-the-art and new trends. *International Journal of Low-Carbon Technologies* **7**, 288-302 (2012).
9. Wang Y, *et al.* Heterogeneous wettability and radiative cooling for efficient deliquescent sorbents-based atmospheric water harvesting. *Cell Reports Physical Science* **3**, 100879-100879 (2022).
10. Li R, *et al.* Hybrid water vapor sorbent design with pollution shielding properties: extracting clean water from polluted bulk water sources. *Journal of Materials Chemistry A* **9**, 14731-14740 (2021).
11. Li Y-T, Chen H, Deng R, Wu M-B, Yang H-C, B. Darling S. Sandwich-Structured Photothermal Wood for Durable Moisture Harvesting and Pumping. *ACS Applied Materials & Interfaces* **13**, 33713-33721 (2021).

12. Shan H, *et al.* High-yield solar-driven atmospheric water harvesting with ultra-high salt content composites encapsulated in porous membrane. *Cell Reports Physical Science* **2**, 100664-100664 (2021).
13. Roy PK, Legchenkova I, Shoval S, Bormashenko E. Interfacial Crystallization within Janus Saline Marbles. *The Journal of Physical Chemistry C* **125**, 1414-1420 (2021).

Reviewers' Comments:

Reviewer #3:

Remarks to the Author:

The work from the authors is now clear to the reviewer. My major concerns on the research have been clarified. I hope the authors can take into consideration my two comments below when working on the final version of the manuscript.

I thank the authors for opening the capsule and make material diagnostics; it a relief for my stability concerns.

Operation: The support material is now more clear in the operation. I suggest the operation method in Fig. R1d should be in the manuscript, not in the support material, along with a sentence similar to the one authors write in their response "his method avoids conducting the inefficient sorption process under low humidity during the day, thus guaranteeing high water uptake of each sorbent as well as high water release yield in a single diurnal cycle. This is an important reason why our work realizes an exceptional water production yield, and together with the advanced thermal design, making our work distinct from the previous work".

LDF: the good fit of the LDF model is not surprising since the LDF model is an empirical kinetic correlation. In most of published literature, the effect of the heat if adsorption on kinetics is often overshadowed by the delays in the experimental apparatus used to measure kinetics (e.g. pressure wave propagation can be slower than kinetics in some materials, sensors lag time slower than material kinetics, heat transfer not at the adequate rate, etc...). The result is that the experimenter sees a time to reach equilibrium longer than the actual kinetics and having exponential trend (the trend assumed by the LDF model). However, while the adsorption times are so long that equilibrium can be assumed (kinetics should not play any significant role), operational desorption times last usually 1-2 hour and kinetics can become more important, with its effects requiring an assessment more careful than what LDF can provide.

Reviewer #3 (Remarks to the Author):

The work from the authors is now clear to the reviewer. My major concerns on the research have been clarified. I hope the authors can take into consideration my two comments below when working on the final version of the manuscript. I thank the authors for opening the capsule and make material diagnostics; it a relief for my stability concerns.

Response:

Thank you for your comments that helped us improve the quality of our work. The two comments were taken seriously into consideration in our final manuscript, and our responses are provided below.

Operation: The support material is now more clear in the operation. I suggest the operation method in Fig. R1d should be in the manuscript, not in the support material, along with a sentence similar to the one authors write in their response “his method avoids conducting the inefficient sorption process under low humidity during the day, thus guaranteeing high water uptake of each sorbent as well as high water release yield in a single diurnal cycle. This is an important reason why our work realizes an exceptional water production yield, and together with the advanced thermal design, making our work distinct from the previous work”.

Response:

Thank you for your comments. We carefully considered your suggestion and modified the manuscript. The figure that represents the proposed operation strategy has been included in the main text as the Figure 4a. The related discussion has been added as well.

Changes made:

Introduction Part: “In this work, we introduce a portable and modularized water harvester with scalable, low-cost, and lightweight LiCl-based hygroscopic composite (Li-SHC) sorbents to realize a hundred-gram water production yield in a real semi-arid climate. Li-SHC shows excellent equilibrium water uptake and optimized sorption dynamics, achieving 1.18, 1.79, and 2.93 g g⁻¹ at 15%, 30%, and 60% RH within night-time 12 hours. More importantly, the gaps between material-level water uptake and real device-level water production remains. Considering the mismatch between relatively low water sorption and fast desorption rates, a batch-processed

operation strategy was proposed in which multiple pieces of sorbents simultaneously capture water vapor to make full use of the nighttime high RH and release them alternately to maintain the high desorption rate during the entire daytime. The combination guarantees high water uptake of each piece of the sorbent as well as high water production yield in a single diurnal cycle. Together with the device-level advanced thermal designs, ...”

Discussion about Figure 4a: “The operation mode of the most reported SAWH devices is quite simple – single water capture-release cycle in a diurnal cycle, resulting in the inefficient water production, largely because of the apparent mismatch between the sorption and desorption dynamics. Such an operation strategy leads to inefficient utilization of solar energy and consequently lowers the corresponding water yield per 24 hours due to the fast desorption rate with a depleted source of sorbed water (Supplementary Fig. 28a). Many research studies attempted to solve this mismatch by semi-continuous or continuous AWH cycles, but this means that the water capture cycles have to take place during the daytime at low RH environments (Supplementary Fig. 28b)^{1, 2, 3}. However, the high RH brought in the nighttime due to diurnal air temperature variation in the semi-arid climates, is more beneficial for the water sorption. To bridge this gap, a distinctive operation strategy, in which the multiple pieces of sorbents are simultaneously exposed to the ambient with high RH in the nighttime to absorb water vapor, then batch-processed alternately for water release during the daytime, is proposed to make full use of both the nighttime high RH environment and maintain high desorption rates throughout the day (Fig. 4a). In practical device operation, this operation strategy requires the portability, adaptability, and stability of both hygroscopic materials and water harvesters. The lightweight sorbent Li-SHC produced with low-cost and sustainable raw materials, facile and easily scaled-up fabrication procedures show excellent water sorption performance, adaptability, and stability, which allows fast deployment into practical applications (Supplementary Information Section 5). Additionally, the portable and cheap water harvester can be easily disassembled, re-assembled, and carried by a person. These features make it possible to realize ultra-high freshwater production yield in a diurnal cycle in real arid scenarios.”

LDF: the good fit of the LDF model is not surprising since the LDF model is an empirical kinetic correlation. In most of published literature, the effect of the heat of adsorption on kinetics is often overshadowed by the delays in the experimental apparatus used to measure kinetics (e.g. pressure wave propagation can be slower than kinetics in some materials, sensors lag time slower than material kinetics, heat transfer not at the adequate rate, etc...). The result is that the experimenter sees a time to reach equilibrium longer than the actual kinetics and having exponential trend (the trend assumed by the LDF model). However, while the adsorption times are so long that equilibrium can be assumed (kinetics should not play any significant role),

operational desorption times last usually 1-2 hour and kinetics can become more important, with its effects requiring an assessment more careful than what LDF can provide.

Response:

Thank you for your comments. We agree with your views about the effect of the sorption heat on kinetics. Non-isothermal effects caused by sorption heat can influence mass transfer, especially at the micro-scale and in seconds. The inclusion of the heat balance to account for non-isothermal kinetics provides a more accurate process description but imposes a significant increase in complexity [*Ind. Eng. Chem. Res.* 2021, 60, 31, 11812–11823]. From this point of view, the LDF model can provide a mathematically simple but adequate description of the dynamic sorption process, and the empirical fitting diffusion parameter that is obtained directly from the uptake data is illustrated as a sensitive parameter in describing the properties of the mass-transfer zones [doi.org/10.1039/F19837900785].

In our work, the LDF model is used on a 12-hour time scale for evaluating the effect of salt content on sorption dynamics to obtain an optimized salt content. We focused on the dynamic sorption characteristics within 12 hours, as we are trying to obtain an optimized maximum water uptake out of the limited 12-hour nighttime with high relative humidity. The LDF model on such a time scale is sufficient as shown by the fitting precision, because the long 12-hour sorption duration, and less sorption heat, have a nearly negligible contribution on the overall water uptake. In the novel models for powder sorbents and hydrogel sorbents, the parameters are also independent on the temperature. For instance, in the reference [*Langmuir* 2022, 38, 8, 2651–2659], the authors claimed that the temperature dependence of parameters such as the diffusion coefficient was neglected in the present case since temperature measurements showed no significant increase during the experiments. Such uniform temperature that resulted from low powder layer thicknesses and constant temperature is maintained in the environmental chamber. In addition, in the reference [*Nano Letter* 2022, 22, 3, 1100 – 1107], Wang's group from MIT developed a sorption model for hygroscopic hydrogels, in which the small temperature difference, of less than 2 °C, for all time, and temperature increase, of less than 10 °C, for all time were found according to the theoretical calculation (Figure R1). They claim that the small temperature rises and temperature gradients in the system have relatively insignificant impacts on the current system.

Figure R1 Simulation results of the temperature profile in the sorbent. Reprint from reference [Nano Letter 2022, 22, 3, 1100 - 1107].

Furthermore, as for the sorbents with relatively lower salt content, the kinetics indeed have not played any significant role as the Reviewer said, owing to the fact that all sorbents reached sorption equilibrium within 12 hours in our test condition. However, the sorbent cannot reach the equilibrium state when the salt content reaches nearly the upper limit (95 wt%), where the LDF model is helpful to evaluate this effect quantitatively.

Meanwhile, we agree with the statement that desorption process which usually lasts only 1-2 hours can be much influenced by kinetics, in which the LDF model may not be suitable anymore. In our case, we do not use this model for desorption but only for evaluating the 12-hour sorption.

Therefore, from this point of view, we prefer to keep this model discussion in the Supplementary Information. We added some discussion about the non-isothermal model and some recently proposed advanced mechanistic models that describe the sorption kinetics of hydrogel-based hygroscopic materials and sorbent powder layers for readers.

Change made:

“The linear driving force (LDF) model assumes isothermal conditions, which is suitable for the conditions without significant temperature increase caused by sorption heat⁴. The inclusion of the heat balance to account for non-isothermal kinetics provides a more accurate process description but imposes a significant increase in complexity. For our 12-hour sorption cases, the LDF model provides a mathematically simple but adequate description of the dynamic sorption process, because temperature measurements showed no significant increase during the experiments, which resulted from low layer thicknesses and constant temperature being maintained in the environmental chamber⁵. Also, the empirical fitting diffusion parameter that was obtained directly from the uptake data is illustrated as a sensitive parameter in describing

the properties of vapor transfer and sorption kinetics⁶. In addition to the LDF model, the kinetic models for hydrogel-based hygroscopic materials and powder sorbent layers were discussed, which considered the volumetric expansion of hydrogel and the porosity of powders, providing frameworks to model the sorption and desorption processes in hygroscopic materials^{5,7}.”

1. Xu J, *et al.* Ultrahigh solar-driven atmospheric water production enabled by scalable rapid-cycling water harvester with vertically aligned nanocomposite sorbent. *Energy & Environmental Science* **14**, 5979-5994 (2021).
2. Li R, Shi Y, Wu M, Hong S, Wang P. Improving atmospheric water production yield: Enabling multiple water harvesting cycles with nano sorbent. *Nano Energy* **67**, 104255 (2020).
3. Hanikel N, *et al.* Rapid Cycling and Exceptional Yield in a Metal-Organic Framework Water Harvester. *ACS Central Science* **5**, 1699-1706 (2019).
4. Peeters R, Verbruggen V, Rongé J, Martens JA. Non-Isothermal Kinetic Model of Water Vapor Adsorption on a Desiccant Bed for Harvesting Water from Atmospheric Air. *Industrial & Engineering Chemistry Research* **60**, 11812-11823 (2021).
5. Legrand U, Girard-Lauriault P-L, Meunier J-L, Boudreault R, Tavares JR. Experimental and Theoretical Assessment of Water Sorbent Kinetics. *Langmuir* **38**, 2651-2659 (2022).
6. Sircar S. Linear-driving-force model for non-isothermal gas adsorption kinetics. *Journal of the Chemical Society, Faraday Transactions 1: Physical Chemistry in Condensed Phases* **79**, 785-796 (1983).
7. Díaz-Marín CD, Zhang L, Lu Z, Alshrah M, Grossman JC, Wang EN. Kinetics of Sorption in Hygroscopic Hydrogels. *Nano Letters* **22**, 1100-1107 (2022).